# Control of membrane barrier during bacterial type-III protein secretion

Svenja Hüsing [1,2], Manuel Halte [1], Ulf van Look[1,2], Alina Guse[1,6], Eric J. C. Gálvez [2], Emmanuelle Charpentier [2], David F. Blair[3], Marc Erhardt [1,2✉] & Thibaud T. Renault[1,2,4,5✉]

Type-III secretion systems (T3SSs) of the bacterial flagellum and the evolutionarily related injectisome are capable of translocating proteins with a remarkable speed of several thousand amino acids per second. Here, we investigate how T3SSs are able to transport proteins at such a high rate while preventing the leakage of small molecules. Our mutational and evolutionary analyses demonstrate that an ensemble of conserved methionine residues at the cytoplasmic side of the T3SS channel create a deformable gasket (M-gasket) around fast-moving substrates undergoing export. The unique physicochemical features of the M-gasket are crucial to preserve the membrane barrier, to accommodate local conformational changes during active secretion, and to maintain stability of the secretion pore in cooperation with a plug domain (R-plug) and a network of salt-bridges. The conservation of the M-gasket, R-plug, and salt-bridge network suggests a universal mechanism by which the membrane integrity is maintained during high-speed protein translocation in all T3SSs.

[1] Institute for Biology—Bacterial Physiology, Humboldt-Universität zu Berlin, Berlin, Germany. [2] Max Planck Unit for the Science of Pathogens, Berlin, Germany. [3] School of Biology, University of Utah, Salt Lake City, UT, USA. [4] CNRS, UMR 5234, Université de Bordeaux, Bordeaux, France. [5] Institut Européen de Chimie et Biologie, Université de Bordeaux, Pessac, France. [6] Present address: Department of Molecular and Cellular Biology, Harvard University, Cambridge, MA, USA. ✉email: marc.erhardt@hu-berlin.de; thibaud.renault@cnrs.fr

**B**acterial type-III secretion is a conserved mechanism that enables assembly of two large nanomachines in the bacterial cell envelope. The bacterial flagellum (assembled by the flagellar type-III secretion system (fT3SS)) is involved in cellular motility through the rotation of a long extracellular filament. Injectisomes (assembled by the virulent T3SS (vT3SS)) are present only in pathogenic Gram-negative bacteria and are used to deliver effector proteins into their eukaryotic hosts. We recently demonstrated that secretion of flagellin via the fT3SS of *Salmonella enterica* occurs through an injection-diffusion mechanism with a remarkably fast initial injection speed—up to tens of thousands amino acids per second[1]—and similar results were reported in *Vibrio alginolyticus*[2]. This high rate of secretion, without equivalent in other pore-based protein channels, raises the question of how membrane gating is preserved. In other words, how does the T3SS secretion pore enable high-speed translocation of proteins with minimal or no leakage of water, ions, and small molecules across the membrane, to preserve membrane gradients and cellular physiology?

Bacteria have evolved several strategies to gate transmembrane channels and transporters. One common feature is the presence of specialized amino acids in the channel of the transporter. For instance, gating in the bacterial Sec translocon is achieved by a stretch of conserved isoleucines in the central constriction of the SecY channel[3], by four glutamines in the MacAB-TolC macrolide efflux pump and eight tyrosines in the Wza polysaccharide transporter[4]. Another approach is the acquisition of gating domains, such as the outward facing plug domain of the SecY translocon[3].

The core of the fT3SS export apparatus (EA) is composed of three proteins, FliP, FliQ, and FliR (SpaP/SctR, SpaQ/SctS, SpaR/SctT in the homologous vT3SS), which form a pseudo-helical secretion pore complex[5]. We previously reported that mutations in a conserved loop of methionines in FliP (M-loop) cause leakage of ions and small molecules through the membrane[6]. However, the mechanism of secretion pore gating and the potential contribution of other components of the core EA remained elusive.

In this work, we perform extensive mutagenesis and functional analyses to systematically investigate the role of FliP/Q/R and of conserved protein features of the secretion pore in maintaining the membrane barrier during type-III secretion. Our results demonstrate that a plug domain in FliR (R-plug) and the M-loop in FliP cooperate to gate the fully assembled T3SS secretion pore. We further show that the ensemble of M-loops of all FliP subunits

in the assembled secretion pore function as a deformable gasket (M-gasket), and that the stability of the secretion pore is maintained by a network of salt bridges in FliQ. The composition of the residues forming the M-gasket required a unique combination of physicochemical features that is found in methionines in order to support both the gating function and the conformational changes that are required for protein substrate secretion. Finally, we demonstrate that gating of the EA is needed during active type-III protein secretion and that mutations compromising the membrane barrier are detrimental to bacterial fitness.

## Results

**The T3SS secretion pore harbors several domains that contribute to seal the secretion channel.** We previously demonstrated that FliP alone can form a pore under over-expression conditions[6]. Accordingly, purified FliP oligomers displayed a decreased central density by electron microscopy reminiscent of a pore[7,8] (Fig. S1a). Recent structural studies have revealed that the functional EA is actually a hetero-oligomer, composed of FliP/Q/R with a 5:4:1 stoichiometry, and forms a helical secretion pore complex[5] (Fig. S1b). An interesting feature of the T3SS pore is its organization in a two-level hetero-oligomer with a pseudo hexameric symmetry. The periplasm-facing top level is composed of five FliP subunits and of the N-terminus of FliR, while the bottom level is composed of four FliQ subunits, the C-terminus of FliR and one FliP subunit (P5) that is common to both levels (Fig. 1a). This peculiar organization is enabled by a striking structural homology in two regions of FliP and FliR (residues 64–100/18–55 and 173–245/71–155), and between FliQ and the C-terminal part of FliR (residues 167–255) (Fig. 1c). Interestingly, the strong structural homology between FliP and FliR is disrupted in the region that faces the central channel, between two α-helices. While FliP features a short, highly conserved loop of three methionines (residues 209–211)—for which we previously reported evidence that it acts as a gasket to close the T3SS channel[6]—FliR exposes instead a bulky domain (residues 106–120) that was hypothesized to act as a plug to seal the channel[5] (Fig. 1b, c).

Our previous work and structural data from others suggest that the T3SS EA has evolved specialized domains to preserve the membrane barrier. Accordingly, we first investigated the individual roles of FliP, FliQ, and FliR, and their cooperation in maintaining the membrane barrier during assembly and substrate export via the fT3SS using a combination of mutagenesis approaches and functional assays (Fig. S1c).

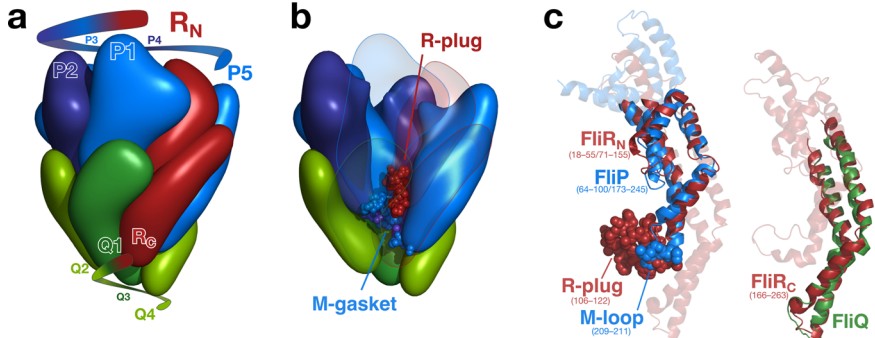

**Fig. 1 Organization of the fT3SS pore. a** The export gate is composed of FliP/Q/R in a 5:4:1 stoichiometry and is organized in a corkscrew-like helical structure. The R subunit has structural homology that is split between P ($R_N$) and Q ($R_C$) (see **c**). **b** The FliR plug (R-plug) and the M-gasket formed of 5 FliP M-loops are located in the core of the pore, sealing it in the closed conformation. **c** The N-terminal part of FliR exhibits a strong structural homology with FliP, with the exception of a bulky domain (residues 106–122) in place of the FliP M-gasket. The C-terminus of FliR is similar to FliQ on its full length. The structures in (**a**) and (**b**) are based on the 6f2d PDB entry. The isosurface at downscaled resolution (8 Å) was computed to better illustrate the global shape and organization of the subunits.

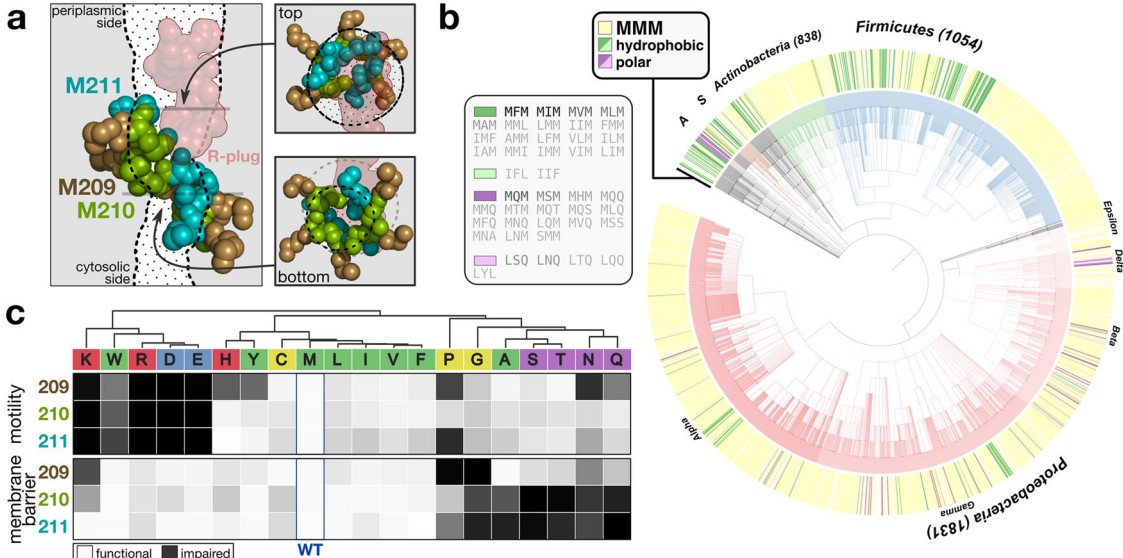

**Fig. 2 Conserved methionines in the PQR pore play a crucial role for T3SS function and gating of the pore. a** Positions of the conserved M-loops that form the M-gasket relative to the channel of PQR pore. The channel was computed using Caver 3 from the 6f2d PDB structure devoid of the M-loops and R-plug residues to model pore opening, and cross-sections were represented as dotted lines and surfaces. The methionine residues are color coded according to their position in the primary FliP sequence. **b** Conservation of the M-loop in flagellated bacteria. The outer ring of the tree is colored according to the M-loop motif. The most conserved MMM motif is colored in yellow, other hydrophobic motifs in green, and polar motifs in purple (with a lighter color when the motif contains no methionine). The color in the central part of the tree represent the major represented phyla: Proteobacteria (decreasing shades of red corresponding to the alpha, gamma, beta, delta, and epsilon classes), Firmicutes (blue), Actinobacteria (green), Spirochetes (S, brown), Acidobacteria (A, gray), and other phyla in black. The alternative motifs are listed on the left of the tree and shaded according to their frequency from black (MFM, 125 occurrences) to light gray (1 occurrence). **c** Clustermap of the motility and membrane barrier function scores for all combinations of single-amino acid substitutions at the three methionine positions. The function scores are defined as the relative motility in soft agar compared to wild type, and as the extent of leakage relative to the mutant with the strongest membrane barrier defect, respectively. The underlying data are presented in Fig. S2e–f. Amino acids are colored according to classical physicochemical groups (red: positively charged, blue: negatively charged, purple: polar, green: hydrophobic, yellow: special).

Briefly, we first quantified the motility of mutants using soft agar swim plates as a proxy for functional flagellar assembly[9]. Second, we quantified secretion of the early fT3SS substrate FlgE (hook protein) fused to the β-lactamase TEM-1 (*bla*) as a proxy of the protein secretion capability of the fT3SS[10]. Third, we quantified gene expression from a flagellar class III promoter using a transcriptional *lac* operon fusion to *fljB* as a reporter of hook-basal-body completion (gene expression from class III promoters is dependent on secretion of the anti-σ factor FlgM after hook completion)[11]. Lastly, we quantified leakage of small molecules through the fT3SS secretion pore by analyzing the water movement kinetics across the membrane as described previously[6].

**A highly conserved methionine loop in FliP plays a dual role to support channel opening while gating the pore.** We previously reported that FliP can form a pore in the inner membrane, both by itself and in combination with other flagellar proteins (now identified to be FliQ and FliR), and described a loop of three methionines (M-loop, residues 209–211 in *S. enterica* FliP) that appeared to play a role in membrane gating by forming an inter-molecular structure that we named the M-gasket. Deletion of a single methionine in the M-loop or mutagenesis of the triplet to alanines caused a drastic increase in membrane permeability[6]. Our initial results suggested that the length of the amino acid side chain was important for the function of the M-gasket. Thereafter, several structural studies have confirmed that the M-loop is indeed exposed to the channel in both fT3SS and vT3SS[5,12,13]. In the FliPQR secretion pore, five copies of FliP are present, which results in a local pool of 15 methionine residues that are spread

over about 20 Å at the opening of the channel (Fig. 2a). This high local enrichment in methionines in combination with the strong conservation of the motif across distant bacterial species (Fig. 2b) raises the question whether methionines have one or more specific properties that could participate in the gating of the T3SS secretion pore.

When we started this study, the structure of the T3SS pore was unknown. Thus, we first sought to identify the biochemical properties required for this putative gating function. We performed a systematic replacement of each of the three methionine residues to all 19 other amino acids in FliP, which was expressed from its native chromosomal location. All mutant FliP proteins were expressed and found in the membrane fraction in levels similar to the WT, suggesting that the majority of single-amino acid substitutions did not affect FliP stability or complex formation in the inner membrane (Fig. S2a). BN-PAGE analysis confirmed that charged residues, for example, caused some degree of EA instability (Fig. S2b). This was further supported by analyzing the efficiency of fT3SS assembly in the membrane using a fluorescent variant of FlhA as a reporter[14] (Fig. S2c). As described above, we then performed three complementary assays to determine the impact of the single-amino acid mutations in the M-gasket on the function of the T3SS (see Fig. S1c for a detailed description of the assays).

Most single substitutions in the M-gasket allowed flagellar assembly and bacterial motility (Fig. S2d–e). Strictly charged residues at any position almost fully abrogated motility. Substitutions to histidine, however, only affected motility on the first methionine position, probably as histidines are mostly unprotonated at cytosolic pH thus greatly reducing their hydrophilicity. Interestingly, a genetic screen for motility

suppressors of non-motile single-charged mutants led to the isolation of a motile M-loop motif mutant, which combined a positive and a negative charge (MER) (Fig. S2g and Table S1). Substitutions to large polar residues (histidine, asparagine, glutamine, and tyrosine) on the first position, rigid prolines on the first and third positions, and bulky tryptophans at all positions also caused a severe loss of motility. The results obtained with the motility assay were confirmed by quantifying secretion of a FlgE-Bla reporter substrate into the periplasm (Fig. S2h).

In order to assess the gating function, we tested membrane permeability in presence of a high concentration (0.5 M) of guanidinium[6]. Ion leakage was reported by the immediate change in optical density ($OD_{600}$). This showed us that preserving the membrane barrier function by the fT3SS pore was mostly dependent on the second and third positions (the residues most exposed in the channel) and that leakage was promoted by substitutions to polar (S, T, N, Q) and small (A, G) residues, and by proline (P) (Fig. S2f).

The motility and membrane barrier function scores were used to cluster mutations with similar effects and revealed three major functional groups (Fig. 2c). The first group contained strictly charged amino acids (R, K, D, E) and bulky tryptophan (W). Substitution to those residues abrogated secretion and motility but had limited impact on membrane permeability, suggesting that the mutants could either be impaired for pore assembly or stability (Fig. S2b), or prevent translocation of the substrates. The second group comprised the polar uncharged amino acids (S, T, N, Q) and the small alanine and glycine residues (A, G). Mutation to those residues moderately affected motility and secretion but strongly increased permeability to small ions through the membrane. This suggested proper assembly of the pore, allowing substrate secretion, but an impaired ability to regulate gating of the pore. Finally, the third group was composed of the hydrophobic medium to large-sized amino acids (V, I, L, M, F, Y), cysteine (C), and histidine (H), all of which had a rather neutral effect on both parameters.

Overall, loss-of-function mutations on the first position (209) affected secretion and motility, but only moderately affected the maintenance of the membrane barrier. This suggested that the first residue primarily plays a structural role, rather than a direct role in maintaining the membrane barrier. Accordingly, the first residue is buried in the PQR complex[5], and substitution to the large residue tryptophan only caused an instability of the EA on the first position (Fig. S2C). On the other side, most mutations of the second (210) and third (211) residues did not affect secretion and motility, with the exception of charged and structure-destabilizing residues (proline, glycine, and tryptophan). The presence of polar residues strongly reduced, however, the ability of the pore to gate small ions. This indicated that those two positions are mostly involved in the gating function, and indeed those residues are exposed in the channel. Furthermore, it suggested that hydrophobicity of the M-loop is an important factor to keep the pore sealed.

**The physicochemical features of methionine are required for the M-gasket to function as a deformable gasket.** Our single substitution approach indicated that hydrophobic amino acids of medium size side chain length were required in the pore-exposed M-gasket of FliP in order for the assembled pore to facilitate both secretion of substrate proteins and preserving the membrane barrier. Interestingly, despite a very strong conservation (Fig. 3a), our results further suggested that methionines are not strictly required at any particular position in the M-loop (Fig. 2b). However, in the context of single substitutions, two methionines

are still present per subunit for one mutated residue. Due to the helical nature of the channel, each residue in the five M-loops has a slightly different position in the M-gasket and different neighbors. Furthermore, it is obvious that conformational changes are required to allow substrate translocation, and this was recently demonstrated in a structure of the vT3SS with a trapped substrate[15]. Thus, it appears reasonable to assume that the global —rather than position-specific—residue composition of the M-gasket is important for fT3SS function.

In line with this hypothesis, we observed that triple substitutions to hydrophobic amino acids other than methionine almost fully abrogated flagellar assembly and motility, suggesting a special role of methionine stereochemistry in this domain that is not restricted to hydrophobicity (Fig. 3b). Single-position mutagenesis had allowed us to identify biochemical properties that were causing defects in T3SS secretion and gating functions when the rest of the M-loop remained unchanged (i.e., one substitution per two methionines). To resolve the properties that are required for those functions in a general context, we generated random triple variants of the M-loop by PCR using mutagenic oligonucleotides with three degenerate codons. The NNS codon allows to obtain full amino acid coverage while minimizing genetic code redundancy, rare, and stop codons. This differed from the single-position mutagenesis experiment in the fact that, in this approach, individual amino acid contributions were challenged in a random, non-optimized context. This setup is thus more likely to reveal important physicochemical requirements of the residues that form the gasket of the secretion pore.

The random triplet variants were either isolated without selection (i.e., allowing any arbitrary combination), or selected to retain some degree of fT3SS function using three different approaches (see Figs. S1c and S3a for a description of the selection methods). As expected, the non-selected variants revealed a homogeneous distribution of all amino acids, matching the frequencies of the NNS codons, while the selected variants demonstrated a clear bias in the frequency of certain residues (Figs. 3c and S3b). Interestingly, the proportion of functional variants (9%) was much lower than expected considering that a functional motif could be any combination of amino acids that did not strongly affect motility (Figs. 2c and S3c). This indicated that the actual combination of residues in the motif is an important parameter to assemble a functional and gated secretion pore.

At first glance, the pattern was very similar to that obtained with the single substitutions: for example, charged amino acids were virtually absent; polar amino acids, proline, glycine, and tryptophan were obtained with lower frequencies, in particular on the first position. There were, however, striking differences compared to the single substitution dataset. For instance, in few occasions charged residues were well tolerated, leading to motility close to that of the WT (Fig. S3d), but only when the opposite charge was also present in the motif (e.g., EGR, CRE, AHE). Those combinations likely result in a null net charge, which confirms that a net charge is detrimental for fT3SS function, as observed in the single substitution dataset. Stringency toward specific amino acids was also substantially more apparent than in the single substitutions experiments. As an example, only medium size hydrophobic amino acids were permissive on the first position, but not large or polar ones. Most importantly, methionines were clearly over-represented in the motifs of the selected variants, which validated that the stereochemistry of methionines has unique features required for the proper function of the M-gasket (Fig. 3c).

To assess the influence of the number of methionines on fT3SS function, we generated variants with all combinations of methionines with a given other amino acid. We chose alanine

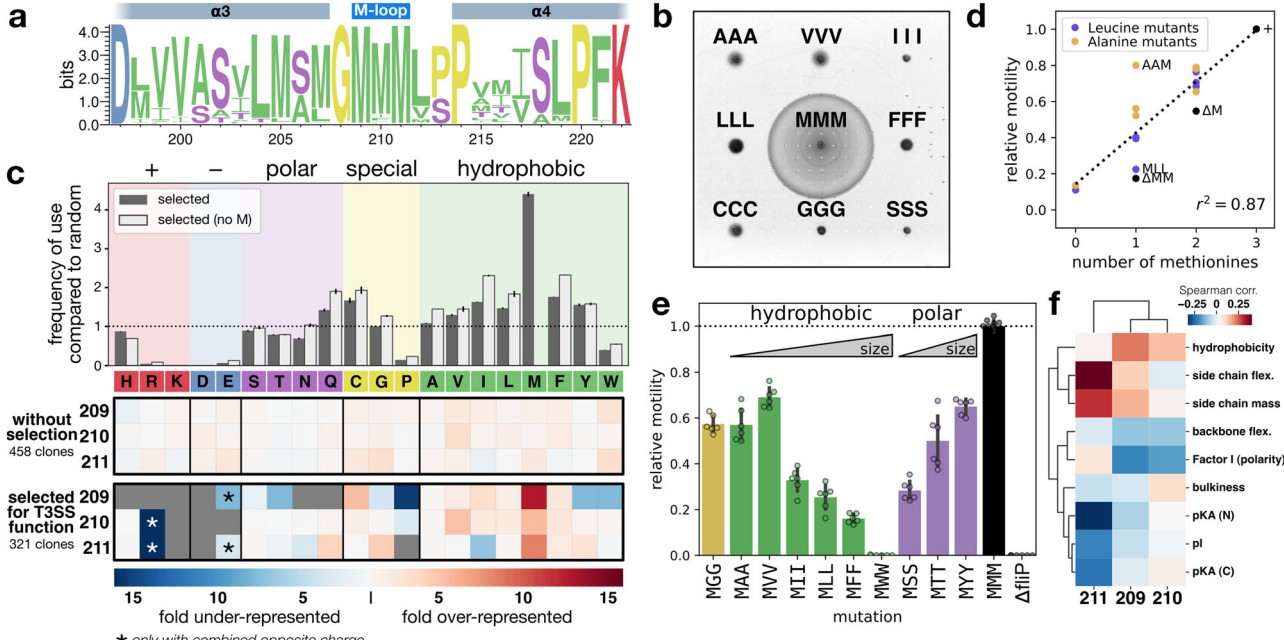

**Fig. 3 Methionine residues display a unique combination of physicochemical features that are required for T3SS function. a** Unlike the FliR plug, methionine residues in the FliP M-loop are highly conserved. From a selection of about 2500 unique sequences of FliP homologs reported in a previous work[7], the first and last methionine residues are 99% conserved and the central one 87%. **b** Motility in soft agar of triple M-loop variants. Unlike single-amino acid substitutions to small and hydrophobic residues that have a limited impact on motility (Fig. 2b), triple substitutions almost completely abrogate fT3SS function. **c** Amino acid usage bias in a set of random triple variants of the FliP M-loop. Variants selected for fT3SS function displayed a strong bias toward specific residues (hydrophobic residues with a net preference for methionines). The bars above the heatmaps represent the frequency of amino acid usage in the motif upon selection relative to the random group, either considering all motifs (dark gray) or excluding motifs containing at least one methionine (light gray) and error bars the standard deviation of the calculated probability. See Figs. S1c and S3a for a description of the selection methods and Fig. S3b for the corresponding individual results. **d** Correlation between the number of methionines in the M-loop and the relative motility in soft agar (see Fig. S3e). **e** Motility in soft agar of various MXX mutants indicates that side chain length of the M-loop residues impacts flagellar function differently depending on the chemical nature of the side chain. $n = 6$ biological replicates, except for MMM ($n = 10$). The bars represent the mean value of the individual measurements and the error bars the standard deviation. **f** Position-specific correlation between fT3SS function (motility score) and the physicochemical features of the M-loop residues. Side chain flexibility, amino acid-size, and hydrophobicity are positively correlated with fT3SS function. In contrast, bulkiness, side chain polarity, and backbone flexibility display a negative correlation. Experiments were performed with at least three biological replicates.

and leucine as those two hydrophobic residues demonstrated rather neutral changes in frequency in our random variants, but have different sizes. Furthermore, mutation from methionine to leucine is the most commonly found natural variation in proteins. If methionines do not play a special role in the M-loop, the motility function should be independent of their number. Overall, fT3SS function was proportional to the number of methionines in the motif (Figs. 3d and S3e). When a single methionine was present in the motif, alanine constituted a better partner than leucine. This emphasized the importance of side chain size and flexibility of the motif, presumably to accommodate channel conformational changes when substrates are traveling through. We thus generated mutants of the two residues that are exposed in the channel (M210, M211) to hydrophobic and polar residues of various side chain size (Fig. 3e). Among substitutions to hydrophobic amino acids, increasing the size of the side chain proportionally reduced motility, yet the opposite effect was observed for polar substitutions. This indicated that while side chain size is an important factor, it is also context dependent. The most illustrative example is perhaps the difference of function between the MFF and MYY mutants. Phenylalanine, a large hydrophobic residue, resulted in a weakly motile phenotype, while tyrosine—which possesses an extra hydroxyl group, and thus has an amphipathic character—resulted in ~70% motility compared to the WT.

Taken together, our data highlight that a non linear, complex combination of physicochemical features is required in this channel-exposed M-gasket to support proper fT3SS function. We thus sought to identify those features in an unbiased manner. For this, we quantified the motility of several hundred unique motifs using a medium throughput assay (Fig. S1c) and computed the correlation with various amino acid physicochemical scales (detailed in "Methods"). In line with our other data, motility was positively correlated with side chain flexibility, intermediate size, and moderate hydrophobicity (Fig. 3f).

We therefore propose that methionine residues combine ideal physicochemical properties in order to create a deformable gasket that accommodates the structural constraints of the T3SS pore, and allows high-speed translocation of substrates, while minimizing or preventing leakage of small molecules. Interestingly, flexibility of the methionine side chain had been identified as a key property in the eukaryotic calmodulin to enable local structural changes and accommodate binding to a large variety of targets[16].

**The FliR plug and FliP M-gasket cooperate to seal the T3SS pore.** The plug of FliR (R-plug) was identified in the previously reported FliPQR structure as a domain composed of large hydrophobic residues and is proposed to act as a gating domain

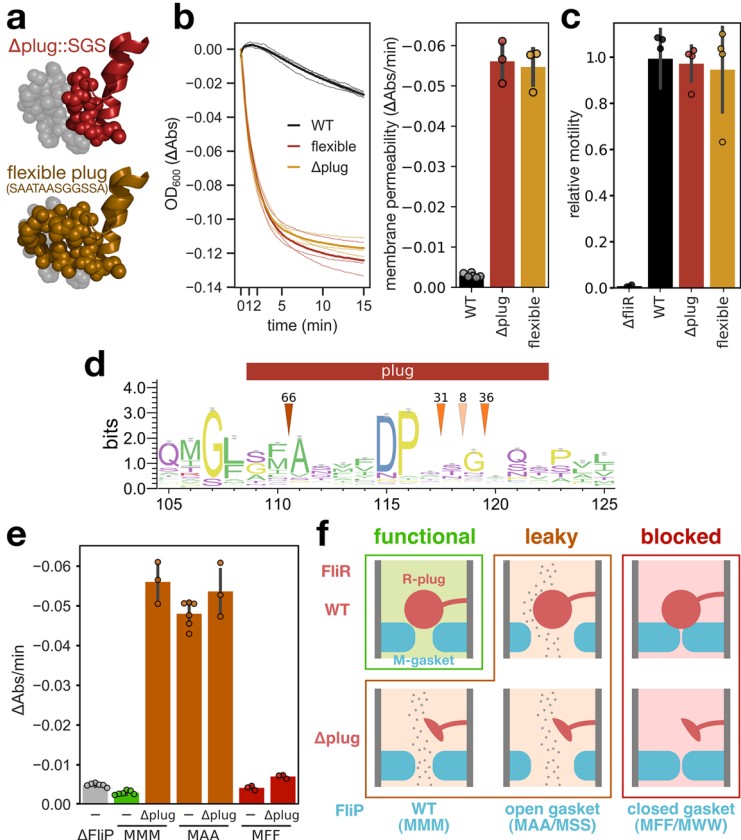

**Fig. 4 The FliR plug cooperates with the M-gasket to seal the T3SS pore and maintain membrane barrier. a** Mutations in the plug domain of FliR: replacement of residues 109–120 (SFATFVDPGSHL) with SGS (top) and substitution of the large amino acids to smaller flexible ones (SAATAASGGSSA, bottom). The WT plug is shown in gray for comparison (PDB: 6f2d). **b** Variation of $OD_{600}$ resulting from the loss of membrane barrier in presence of salt. Both R-plug mutations induce a strong leakage through the fT3SS pore. **c** Flagellar function reported by motility in soft agar. **d** The plug domain is poorly conserved across FliR homologs (numbering according to FliR of *S*. Typhimurium). Arrows indicate the number of homologs with insertions at the pointed location. **e** Leakage through the fT3SS pore for combinations of M-gasket and R-plug mutations. The reported value corresponds to the initial slope of the water movement kinetics of Fig. S4c. Combinations of leaky mutations do not have an additive effect suggesting that the M-gasket and R-plug cooperate in gating the fT3SS pore. Conversely, a mutation in the M-gasket that tightens the channel (MFF) prevents leakage induced by a R-plug mutation. $n = 4$ biological replicates, except for $\Delta fliR$ ($n = 3$). **f** Model of the cooperation between the M-gasket and R-plug. $n = 6$ biological replicates for the WT/MMM, $\Delta fliP$, and MAA strains, $n = 3$ for the other mutants in (**b** and **e**). The bars represent the mean value of the individual measurements and the error bars the standard deviation.

as it occludes the channel in its closed conformation[5], but it remained to be functionally characterized. We therefore performed directed mutagenesis of the plug domain, in order to determine the potential role of the R-plug in gating of the T3SS channel. We generated several mutants that destabilize this domain by either substituting the large or rigid amino acids of the domain (residues 109–120, SFATFVDPGSHL) with smaller flexible residues (SAATAASGGSSA), or by replacing the domain with short flexible serine/glycine loops (SGS, SGSGS, or SGSGSGS) (Fig. 4a). Like for the M-loop mutants, we monitored motility and leakage through the pore. The plug mutations resulted in a strong permeability to ions across the membrane (Figs. 4b and S4c). Yet, the mutated plugs did not affect flagellar function (Fig. 4c). Despite the loss of membrane barrier, the mutants displayed a motility phenotype similar to that of the WT, even in the presence of high concentrations of ions in the medium (Fig. S4d). This suggested that, unlike the M-gasket, the role of the R-plug is restricted to gating the pore.

The observation that mutations in the R-plug affect gating of the pore, but not flagellar assembly and function suggests that tight membrane gating is not strictly required for protein translocation via the T3SS per se, but is rather important for

the general physiology of the cell. In addition, the poor sequence conservation, and the high phylogenetic diversity (Fig. 4d) support the idea that the plug in FliR may have acquired a specialized role to seal the pore in its closed conformation.

The close proximity between the FliP M-gasket and the FliR plug (~3.5 Å) raises the possibility that the two domains cooperate in gating of the channel (Fig. S4a). If the M-gasket and the R-plug act as two independent gating domains, then it would be expected that a combination of both mutations is required to cause the greatest defect. In contrast, if the two domains act in cooperation, failure of either of them would be sufficient to impair the gating function. To address this issue, we measured the leakage of several combination of M-loop mutants in presence of a deletion of the R-plug.

The most leaky mutants of the M-loop and mutants of the FliR plug induced a comparable membrane permeability. We observed, however, no additive effect when both mutations were combined (Figs. 4e, f and S4c). This suggested that the two domains do not have a redundant function but rather are simultaneously required to gate the fT3SS pore, probably acting in cooperation. To further validate this observation, we introduced mutations to bulky amino acids in the M-loop to

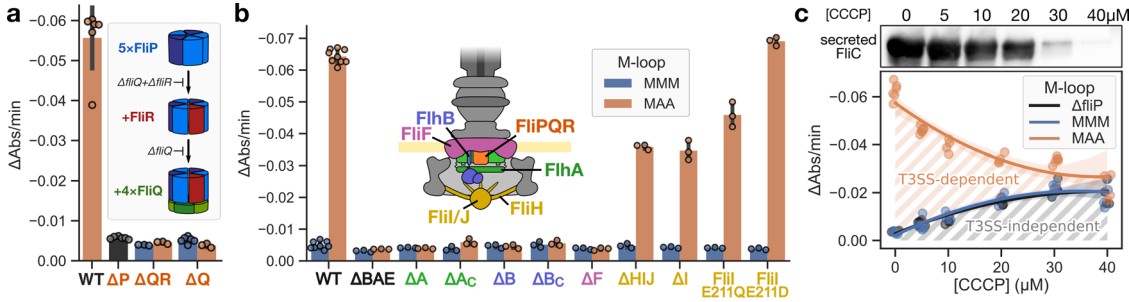

**Fig. 5 fT3SS pore gating is only required when the EA is assembled and active.** We used a leaky M-loop mutant (MAA, in brown) to identify the stages in which the fT3SS pore is open and requires gating. FliP/Q/R assembly intermediates (**a**) that do not form an open pore[7] and mutants deficient in fT3SS function (**b**) do not leak small molecules. Mutants of the FliHIJ ATPase complex retain some degree of fT3SS function[18–20]. ΔFlhA_C/ΔFlhB_C are mutants of the C-terminal domains, which do not play a structural role. **c** Cells were treated with CCCP to alter the PMF and gradually decrease secretion. Top: secreted proteins were separated by SDS-PAGE and flagellin was detected by Western blot. As expected, increasing concentrations of CCCP cause a gradual decrease in protein secretion. Bottom: dependence of MAA-induced leakage on the level of secretion. CCCP was added to the cells immediately before monitoring the kinetics of water leakage. The intensity of leakage caused by the MAA mutation is inversely proportional to the concentration of CCCP up to a point where it reaches that of the WT and ΔfliP controls. NB high concentrations of CCCP induce fT3SS-independent leakage. The error band represents the 95% confidence interval. $n = 3$ biological replicates for all strains of (**a** and **b**), except ΔfliP/ΔfliQ/MAA in (**a**) ($n = 6$) and WT (both MMM/MAA) in (**b**) ($n = 8/n = 9$, respectively). The bars represent the mean value of the individual measurements and error bars the standard deviation.

tighten the channel (MFF and MWW) and combined it with the deletion mutant of the FliR plug. Both M-loop mutations prevented leakage through the channel in the FliR plug mutant background (Figs. 4e, f and S4c).

**Intermediates of fT3SS assembly do not require gating.** The fT3SS core complex assembles sequentially. FliP oligomerizes first with the help of the integral membrane chaperone FliO, FliR is then recruited, and finally FliQ assembles at the bottom of the structure[7,8]. Once the pore complex is assembled further subunits such as FlhB, FlhA, and FliF continue assembling to form the EA. We previously observed that plasmid over-expression of FliP resulted in pore formation that causes a detectable loss of membrane barrier function upon removal of one of the methionines of the M-loop[6,7]. We thus wondered if this represented the physiological situation and, more generally, if intermediates of assembly required gating.

For this, we introduced a leaky mutation (MAA) into the M-loop of various intermediates of assembly of the pore (ΔfliQ/R mutants) or of the EA (ΔflhA/B, ΔfliF mutants). While the fully assembled EA was leaky in presence of the MAA mutation in FliP, this was not the case for the intermediates of assembly (Figs. 5a, b and S5). This indicated that the EA assembly intermediates are not able to form an open pore in the membrane.

**fT3SS gating is only required during active secretion.** Interestingly, the MAA mutation was also not causing leakage in deletion mutants of the soluble C-terminal domains of FlhA and FlhB, which still assemble the EA, but are deficient in substrate secretion[17]. This suggested that leakage might only occur during active secretion.

It has previously been shown that the fT3SS is intrinsically a proton motive force (PMF)-dependent protein exporter[18,19], while the associated FliH/I/J ATPase complex contributes to efficient substrate secretion and the fT3SS retains some degree of function also in ATPase mutants[18–20]. We first investigated leakage of the MAA mutant in a background deleted for the FliH/I/J ATPase complex (Fig. 5b). We analyzed these mutants because we reasoned that in the absence of the ATPase complex or in ATPase point mutants that are (partially) deficient in ATP hydrolysis, the EA would still properly assemble, but protein

secretion via the fT3SS would be reduced to various degrees compared to the WT. As shown in Fig. 5b, both in deletion mutants of the ATPase, as well as in ATPase point mutants, the MAA mutation in FliP resulted in leakage of small molecules across the cytoplasmic membrane. This observation is particularly interesting for the ATPase point mutants. The FliI E211D mutant has reduced ATPase activity but secretes most substrates efficiently. In contrast, the FliI E211Q mutant has no ATPase activity and greatly reduced secretion[21]. Accordingly, leakage in the FliI E211D background was similar to that of the WT while the FliI E211Q mutant displayed an intermediate phenotype.

We next employed the ionophore and uncoupler carbonyl cyanide m-chlorophenyl hydrazone to disrupt active protein secretion via the fT3SS[19]. As shown in Fig. 5c, leakage of small molecules in the FliP MAA mutant decreased in a dose-dependent manner. This result suggested that the primary role of the M-gasket is to preserve the membrane barrier during active protein secretion via the fT3SS.

**Systematic suppressor analysis of the M-loop reveals residues important for fT3SS secretion pore function and gating.** Both the methionine loop in FliP and the plug domain in FliR are involved in gating of the fT3SS channel. While the plug in FliR appears to be dedicated to gating the channel, the role of the M-gasket is likely a trade-off between gating the channel and accommodating deformation of the gasket during secretion pore opening and substrate translocation. Our single substitutions experiment indicated that only the second and third methionines of the M-loop (M210, M211) are involved in gating the channel, while the first one (M209), which is buried in the protein structure, is rather involved in maintaining the stability of the complex (Fig. 2c). Accordingly, the poorly functional AAA mutant assembles less flagella (Fig. S6a) and is therefore less leaky than the single and double alanine mutants (Fig. S3g). This idea was further reinforced by the analysis of motile suppressors from the homogeneous triple mutants of the M-loop (Fig. 6a, b). The almost non-motile triple M-loop mutants were inoculated in soft agar motility plates and incubated until flares of motile bacteria were observed, which indicated the acquisition of a gain-of-function mutation. Interestingly, motile suppressors of small residue M-loop mutants (GGG, AAA, SSS) were easily acquired and led exclusively to extragenic mutations in well characterized

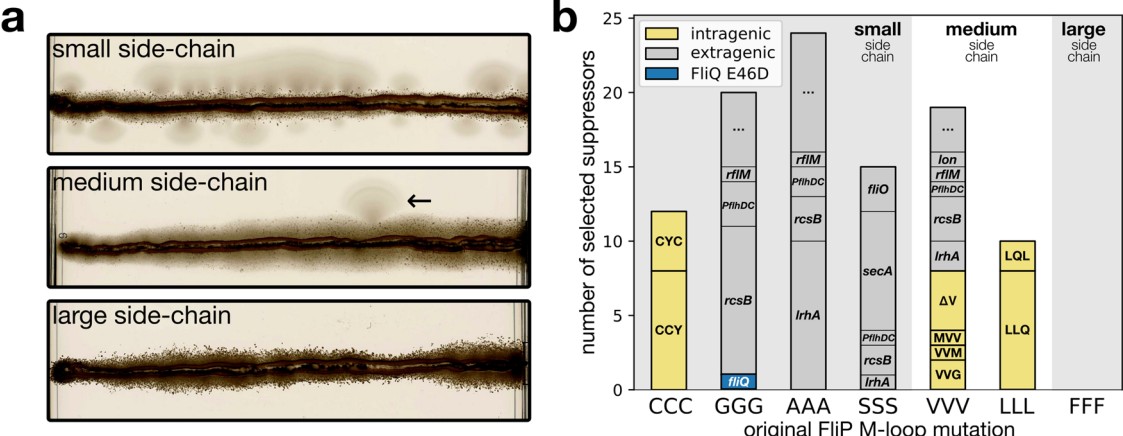

**Fig. 6 Suppressor analysis of the M-loop. a** Weakly motile mutants from Fig. 3b were inoculated into soft agar and monitored for several days until "flares" of motile suppressors were obtained. M-loop mutants to small amino acids had a high rate of acquisition of suppressor mutations, medium size amino acids gave isolated suppressors. We never obtained suppressors for the mutant to large FFF residues. **b** The nature of the mutations identified in the gain of function suppressors correlates with the size of the M-loop residues. Small size M-loop suppressors (GGG, AAA, SSS) acquired almost exclusively mutations in regulatory genes of flagellar assembly. The large size LLL variant required mutations within the M-loop to restore motility function. Medium size variants (VVV) acquired either inter- and extragenic mutations. The CCC mutant behaved like a large side chain variant, which we interpret as a possible channel constriction due to inter-cysteine di-sulfide bond formation. Interestingly, M-loop mutations consisted in substitutions to amino acids with different physicochemical properties (polar vs hydrophobic, small vs large), acquisitions of methionines or deletion of one residue. This indicated that the CCC, VVV, and LLL mutants are likely reducing channel size, impairing substrate secretion, while the small, highly flexible GGG, AAA, and SSS motifs are probably leading to unstable protein complexes. Gain of function mutations identified in well-known regulators of flagellar assembly are represented in gray. Accordingly, the polar M-loop variant SSS gave rise to mutations in *fliO* and *secA*, both of which are directly implicated in FliP assembly. "…" indicates that no obvious suppressor candidate was identified.

flagellar genetic regulators that are known to induce an over-expression of flagellar genes (Fig. 6b and Table S2). We hypothesize that over-expression of flagellar genes bypasses assembly defects of these triple M-loop mutants, similar to suppressor mutations bypassing the requirement of the fT3SS associated ATPase[20] or flagellar C-ring[22]. Conversely, motile suppressors of M-loop mutants consisting of larger and more rigid amino acids almost systematically affected the M-loop to incorporate smaller, more flexible or more polar residues, with the exception of the FFF variant for which we never isolated suppressors (Fig. 6b). Intriguingly, the GGG variant gave rise to a suppressor bearing a FliQ E46D mutation. This gave us a first indication that FliQ could impact the function of the M-gasket.

**Salt bridges in FliQ stabilize the FliPQR complex.** FliQ is located on the cytosolic side of the FliPQR pore. It assembles last into the FliPQR complex[7]. While the protein is not required to form the initial, stable, FliPR complex, FliQ is essential to complete the assembly of the fT3SS secretion pore. FliQ possesses a relatively conserved methionine (M47) that is in close proximity to the FliP M-gasket (4.7 ± 0.9 Å). We initially hypothesized that this residue could contribute to the pool of methionines in the M-gasket and therefore participate in gating the pore. However, mutations of FliQ M47 were well tolerated (even to charged residues) and had only a limited impact on motility and on gating of the pore (Fig. S6b, c). This suggested that the M47 was not required for M-gasket formation and fT3SS function but might rather participate in fine tuning of gating.

The four FliQ subunits form α-helical hairpins with a kink in their center. Their tight interaction is supported by inter-molecular salt bridges involving the residues E46 and K54, and it was hypothesized that they are important for the stability of the complex[5]. Accordingly, we found that mutations of those residues to alanine almost fully impaired motility (Fig. S6b), which confirmed that a tight interaction between the FliQ subunits is required for fT3SS function.

Before the structure of the FliP/Q/R complex was known, we had performed tryptophan replacements in the two α-helices of FliQ to destabilize their structure and identify important residues and potential interfaces of interaction. We obtained two mutations that fully abrogated motility: A28W and L29W (Fig. S6d), presumably by preventing EA assembly or substrate translocation. Interestingly, the L29W variant spontaneously gave rise to a motile suppressor consisting of a single methionine deletion in the M-loop of FliP. This result was the second indication in our data suggesting a possible interplay between FliQ subunit packing and the function of the M-gasket. We therefore propose that FliQ might act as a clamp to maintain the stability of the secretion pore, and eventually indirectly contribute to gating of the secretion pore (Fig. S6e). Given the small change resulting from the E to D mutation in the FliQ E46D suppressor mutant, we hypothesized that the smaller aspartic acid would pack the structure even further, and that this mutation should decrease fT3SS function in a WT background of the FliP M-loop. As expected, in this background the E46D mutation resulted in a slight decrease in motility (~20%) (Fig. 7a).

To further assess the connection between FliQ packing and the function of the FliP M-gasket, we combined M-loop variants with the FliQ E46D mutation. Depending on the nature of the M-loop mutants, the FliQ E46D mutation had antagonistic effects. M-loop mutations to large amino acids (FFF, LLL, VVV) behaved like WT FliP: motility was further decreased and no change in leakage was detected. Consistent with our hypothesis, the FliQ E46D mutation had the opposite effect in combination with M-loop mutations to small amino acids (GGG, AAA, SSS): fT3SS function was improved, and leakage was increased (Figs. 7a, b and S6f), which we attribute to enhanced stability of the PQR complex. Accordingly, the FliQ E46D mutation improved the flagellation of the AAA M-loop variant (Figs. 7c and S6a).

The FliQ residues involved in forming the inter-subunit salt bridges are conserved in all currently documented homologs. The vast majority utilize a glutamic acid (E), often in combination

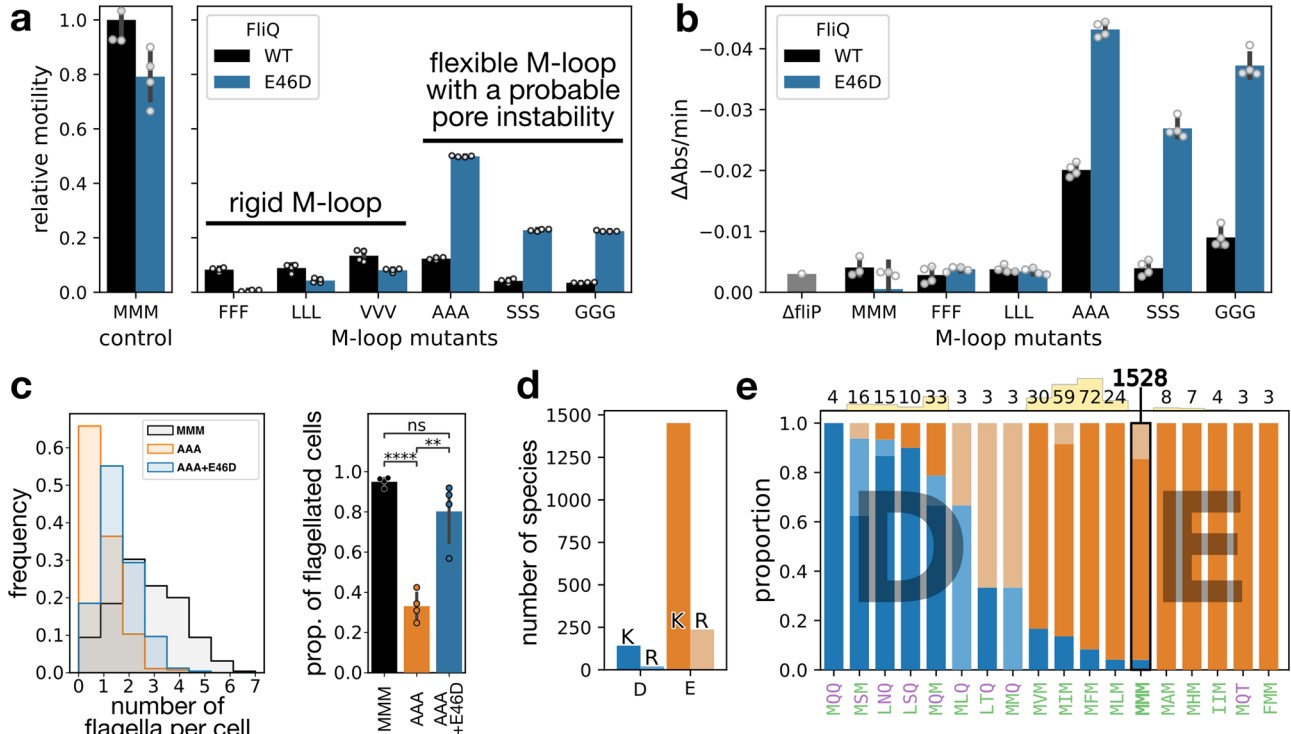

**Fig. 7 FliQ stabilizes the fT3SS secretion pore through a network of salt bridges. a** Effect of the FliQ E46D mutation on the motility in soft agar of poorly motile M-loop mutants. FliQ E46D has a positive effect on motility of flexible M-loop variants, and a negative effect for the WT and rigid M-loop variants. $n = 4$ biologically independent samples. **b** The stabilizing effect of the FliQ E46D mutation enhances leakage induced by the small and flexible M-loop mutants. $n = 4$ biologically independent samples, except for MMM ($n = 3$) and $\Delta fliP$ ($n = 1$). **c** The FliQ E46D mutation improves the flagellation of the defective AAA M-loop variant, both in terms of number of flagellar filaments per cell (left) and of proportion of flagellated bacteria (right). The statistical test is a two-sided $t$-test with Bonferroni correction for multiple testing ($^{ns}p = 0.35$; $^{**}p = 5.2 \cdot 10^{-3}$; $^{****}p = 1.2 \cdot 10^{-5}$). **d** The majority of FliQ homologs use a glutamic acid (E) in their inter-molecular salt bridge combined with a lysine (K). **e** The small proportion of FliQ that have an aspartic acid (D) are strongly correlated with M-loop variants possessing a polar residue. The number of occurrences of each M-loop variant is indicated on the top of the graph and represented as a small yellow bar for comparison (except for the WT due to scale). Hydrophobic and polar residues of the M-loop are colored in green and purple, respectively. Salt bridges with lysine (K) and arginine (R) are colored in dark and light shade, respectively. Only motifs found in more than two occurrences have been included. Experiments were performed with at least three biological replicates. The bars represent the mean value of the individual measurements and the error bars the standard deviation.

with a lysine (K), while the presence of an aspartic acid (D) remains rare (Fig. 7d). Interestingly, we observed that the use of an aspartic acid correlates with the presence of hydrophobic residues in the M-gasket that are prone to induce ion leakage according to our data from *S. enterica* (Fig. 7e). The exact impact of the presence of hydrophobic residues in the M-loops of these species would need to be investigated, however the observed correlation suggests that the FliQ E46D variant might have co-evolved with the M-loop to maintain the membrane barrier for these M-loop compositions.

In summary, our data indicate that the M-gasket has a dual role in gating the T3SS secretion pore and accommodating conformational changes required for high-speed substrate translocation. The latter function likely requires a deformation of the structure of the M-gasket that challenges the stability of the secretion pore. We observed that FliQ packing impacts the bulk of the M-gasket (and thus, likely the size of the secretion pore) and, reciprocally, that instability triggered by flexible M-loops can be corrected by a reorganization of the inter-FliQ salt bridges network. This indicates that FliQ plays a crucial role in maintaining the stability of the T3SS pore, and indirectly in maintaining the membrane barrier.

**Physiological role of T3SS gating.** Our extensive functional analysis of conserved features of the T3SS EA demonstrates that

gating of the secretion pore is achieved by a combination of several factors: a flexible deformable M-gasket made of the M-loops of five FliP subunits, a bulky plug domain in FliR that cooperates with the M-gasket to seal the pore, and the stabilization of the complex by a network of salt bridges between the four FliQ subunits. Furthermore, our results indicate that the export gate is likely a dynamic structure that supports conformational changes during secretion of substrate proteins (Fig. 8a). Our analysis of FliR plug and FliP M-loop mutants demonstrated that those two domains prevent passage of water and ions across the secretion pore. Loss of membrane barrier results in alteration of bacterial physiology through disruption of the cellular gradients (e.g., the PMF) and unwanted exchange of molecules across the membrane. To assess the importance of T3SS pore gating for bacterial physiology, we therefore compared the fitness of several mutants of the FliP M-loop expressed from their native chromosomal location. Mutations that caused leakage through the pore induced a growth defect (Fig. 8b), highlighting that despite the low number of secretion pores per cell, losing the ability to gate the fT3SS pore is detrimental for fitness.

## Discussion

The type-III protein secretion machinery allows for the assembly of megadalton nanomachines such as the flagellum and the related injectisome in the bacterial cell envelope. The assembly

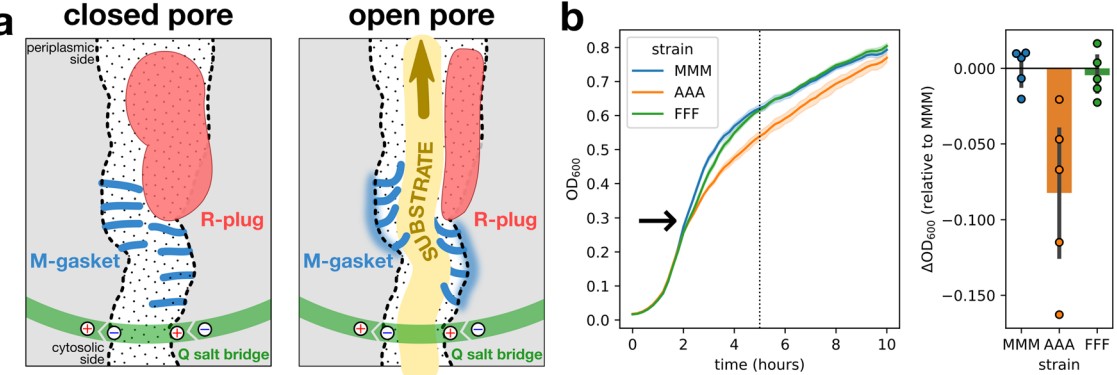

**Fig. 8 The T3SS secretion pore conserves membrane barrier during type-III secretion to maintain fitness. a** Model of fT3SS secretion pore regulation to preserve membrane barrier. In closed conformation, the M-gasket (blue) and the R-plug (red) cooperate to seal the channel and prevent alteration of membrane barrier. Upon substrate translocation (yellow), the M-gasket and R-plug undergo conformational changes to enable opening of the channel. Salt bridges in FliQ (green) are critical to support M-gasket deformation and maintain the secretion pore tightly assembled. **b** The loss of membrane barrier caused from leaky fT3SS secretion pores results in altered fitness. Left: growth of several M-loop variants was followed over time by monitoring the $OD_{600}$. The arrow indicates the growth phase ($OD_{600} = 0.3$) at which expression of the flagellar master regulator $FlhD_4C_2$ starts increasing[35]. Right: the FliP AAA mutant, that is impaired in secretion pore gating, has a growth defect compared to the WT (MMM) and the FFF variant. $n = 5$ biological replicates. The bars represent the mean value of the individual measurements and the error bars the standard deviation. The error bands (left) represents the 95% confidence interval.

process of these complex nanomachines requires the secretion of dozens of protein subunits of various sizes and stoichiometries. Importantly, the T3SSs of the flagellum and the injectisome are capable of translocating substrate proteins across the inner membrane with a remarkable speed of several thousand amino acids per second, which is several orders of magnitude faster than protein export in other pore-based protein channels[1,23]. Furthermore, engineering of this secretion system has demonstrated that a wide range of proteins can be transported as substrates[24]. Accordingly, it appeared reasonable to assume that a complex multi-level gating mechanism has evolved to maintain membrane integrity and to prevent the leakage of water and small molecules during high-speed substrate protein secretion.

The structures of the closed secretion pores of the T3SSs of the flagellum and the injectisome suggested that a plug domain in FliR and a stretch of methionine residues in FliP form constriction points that might contribute to gating of the T3SS[5]. However, functional evidence of how the T3SS is able to facilitate the high-speed transport of substrate proteins while preventing the leakage of small molecules was lacking. In particular, the contribution of the highly conserved stretch of methionines in FliP in maintenance of the membrane barrier during active secretion remained unclear.

In this work, we performed an extensive functional analysis of the structural features of the T3SS secretion pore that were previously hypothesized to contribute to the gating mechanism. Our results demonstrate that the M-gasket made of five repetitions of a conserved loop of methionines in FliP (M-loop) and the R-plug domain in FliR are critical features of the fT3SS secretion pore to preserve the membrane integrity during high-speed protein translocation. As suggested by the previous structures of the closed secretion pore, both the M-gasket and the R-plug seal the channel in its closed conformation, which renders any protein translocation impossible. Thus, conformational changes of both domains are required to enable substrate export. The FliR plug appears to be dedicated to the gating function, and its deletion does not otherwise affect flagellar function. In contrast, alterations in the M-loop frequently cause functional defects, supporting the idea that the M-gasket plays a dual role during T3SS function. It acts both as a flexible gasket to maintain the membrane barrier during substrate protein translocation, and as a hinge to accommodate pore opening and substrate translocation. The convoluted spatial organization of the M-gasket in the channel attributes a unique

position to each of the 15 methionine residues of the M-loops, that is spread over 20 Å in the secretion channel. This suggests that different residues in each M-loop might contribute to the formation of the M-gasket over the different events of pore opening and substrate translocation.

Using extensive targeted and random mutagenesis of the M-loop, we demonstrated that a complex combination of physico-chemical features of residues forming the mouth of the T3SS secretion pore is required to support membrane gating and substrate secretion, and that unlike any other amino acid, methionines ideally fulfill this requirement. Moderate hydrophobicity, a long side chain, and side chain flexibility are likely important to maintain a dynamic interaction between the FliP subunits and to prevent leakage by maintaining a close contact with the secreted substrates while accommodating conformational changes required for their rapid translocation. This immediate proximity with the substrates is likely a reason why a net charge is not tolerated in the channel.

Maintaining membrane barrier during protein secretion is crucial to bacterial fitness. This was shown for the bacterial Sec translocon that involves hydrophobic residues and a plug domain to gate the constricted section of the channel[3]. We report here a similar observation for the T3SS in which a leaky M-gasket induces a growth defect. This is further highlighted by the fact that only a handful of T3SS secretion pores are assembled in the cell envelope. Interestingly, while active secretion reduced leakiness in mutants of the Sec translocon, we observed the opposite in the fT3SS. Leakage caused by a mutation in the M-loop of FliP occurred only in actively secreting fT3SS.

The close structural homology between the three components of the secretion pore makes it tempting to hypothesize that the T3SS secretion pore evolved sequentially from a simpler pore, by specialization of the FliR subunit and of its plug domain. However, the potential function and impact of specific adaptations on secretion pore gating, such as the extra methionine residues in the M-loop of some virulent T3SS homologs, or conversely the existence of rare flagellar homologs with reduced number of methionines in the M-loop remains to be investigated.

## Methods

**Bacteria, plasmids, and media**. Strains used in this study derived from *Salmonella enterica* serovar Typhimurium strain LT2 and are listed in Table S3. Lysogeny

broth (LB) contained 10 g of Bacto-Tryptone (Difco), 5 g of yeast extract and 5 g of NaCl per litre. Soft agar plates used for motility assays contained 10 g of Bacto-Tryptone, 5 g of NaCl and 3.5 g of Bacto-Agar (Difco) per liter.

**Chromosomal modifications**. Chromosomal mutations were obtained using lambda-RED-mediated recombineering[25] and a Kan-SceI cassette as selection marker[26]. Site-specific chromosomal insertion of the Kan-SceI cassette and subsequently the replacement with the desired chromosomal mutation was achieved by inducing the expression of the λ-Red phage genes (gam, bet, exo, encoded on the temperature-sensitive plasmid pWRG730), by heat-shock. Both, the Kan-SceI cassette and the replacement fragments were amplified by PCR with flanking homologous regions of 40 bp to the target region. The desired chromosomal modification was obtained by counter-selection on LB plates containing anhydrotetracycline to induce expression of the meganuclease I-SceI. Recombinant colonies were verified by PCR, followed by sequencing. The primers used in this work are listed in Table S4.

**Random mutagenesis**. Random mutagenesis of the FliP M-loop was achieved by using a degenerate oligonucleotide with a homologous overhang complementary to the sequence immediately upstream of the M-loop (cgacctggtgatcgc-cagcgtattgatggcgttgggg **NNS NNS NNS** gtgccgccagcgac). The **NNS** codon (N = [ACGT]; S = [GC]) allows to obtain all possible amino acids with limited redundancy and avoiding rare and stop codons. Mutagenic fragments were amplified by PCR using the degenerate oligonucleotide together with a reverse primer (gctgataatgaggccggtaa), and were subsequently recombined in the chromosome using the λ-Red procedure. Selection for functional variants was carried out either by selecting for ampicillin-resistant clones in a genetic background containing a substrate fused to the β-lactamase (see "Quantification of substrate secretion") or by picking and subcloning flares of motile bacteria after short-term growth in soft agar. Alternatively, whether variants were able to assemble a functional flagellum was determined using a genetic reporter (see "Genetic reporter of complete flagellum assembly").

**Motility assay in soft agar**. In order to assess the motility of the various *Salmonella* mutants, overnight cultures were inoculated in soft agar plates with a pin tool and incubated at 30 °C. Plates were scanned automatically from within the incubator at regular intervals using the light transmission mode of a Perfection V800 Photo scanner (Epson) controlled by a custom software. Motility was determined from the radius of the rings of cells originating from the points of inoculation (Fig. S1c). Background was subtracted using an image of the plate shortly after inoculation as a reference. Motility rings were detected using ImageJ by applying a threshold (Huang algorithm) and automatic detection of the contours of the rings.

**Isolation of genetic suppressors of motility defects**. Poorly motile or non-motile mutants were inoculated in soft agar and incubated for up to several days. Plates were monitored to detect the apparition of motility flares (see Figs. 6a and S3a as examples of motility flares). Suppressors were isolated by sampling the motility flares with a toothpick. The phenotype of the suppressors was validated by performing a motility assay, and the resulting clones were sequenced.

**Analysis of the physicochemical properties required for the function of the M-loop**. Functional random M-loop mutants that were isolated with the "motility" and "*fljB::lacZ*" methods (described in Fig. S3a) were subjected to a motility assay in soft agar to quantify flagellar function (see Fig. S1c). The correlation between motility and various amino acid scales was calculated and the matrix of correlations was clustered. Spearman correlation was computed here, but similar results are observed using Pearson correlation or distance metrics. The amino acid scales that were used here are the Rose et al. hydrophobicity scale, the Bhaskaran and Ponnuswamy backbone flexibility scale, the Zimmerman et al. bulkiness scale as defined in https://web.expasy.org/protscale/; Factor I is defined in[27]; side chain flexibility is defined in[28]; the mass of the side chain, pKAs, and pI are textbook values.

**Quantification of substrate secretion**. Mutations of interest were transduced in a strain carrying a ΔflgBC mutation and a translational fusion of the hook protein FlgE to the β-lactamase TEM-1 lacking its secretion signal (FlgE-Bla)[29]. This permitted secretion of FlgE-Bla directly into the periplasm and conferred resistance to ampicillin proportionally to the amount of secreted substrate. The ampicillin IC50 value was determined by growing the bacteria in a 96-wells plate with a titration of ampicillin (0–1024 μg/ml) and by fitting a logistic function on the endpoint $OD_{600}$.

**Kinetics of water movement across the inner membrane**. In this assay, the inward flux of water across the inner membrane is quantified by monitoring the drop of absorbance that results from cell swelling. The ground principle of this assay is described in detail in a previous work[6]. Bacteria were grown in LB medium containing 10 g/l NaCl until reaching an $OD_{600}$ of 0.6. Cells were washed twice in 1 × PBS, concentrated to 30 $OD_{600}$/ml, and diluted 1:1 in a high concentration of salt (e.g., 1M guanidinium in 1 × PBS). Absorbance at 600 nm was monitored immediately after addition of the high salt solution using Cytation3 or Synergy H1

plate readers (Biotek), and variation over time reflected the flux of water across the inner membrane. The initial speed of efflux was quantified, after standardization to the OD unit, as the slope of the initial tangent to the curve (ΔAbs/min).

**Genetic reporter of complete flagellum assembly**. Complete assembly of the flagellum was assessed by reporting the genetic switch to flagellar class III genes. For this, a transcriptional fusion of the *fljB* flagellin and the MudA transposon, that carries the β-galactosidase (*lacZ*) gene[30], was used. Bacteria able to assemble the flagellar hook and switch to late substrate expression thus appeared as red colonies on MacConkey agar plates (Difco) containing lactose.

**Immunostaining of flagellar filaments**. To determine the number of flagella on individual bacterial cells, flagellar filaments were labeled with anti-flagellin antibodies coupled to organic fluorochromes. Bacteria were grown till mid-log phase and flagellar filaments were incubated with a 1:1000 dilution of polyclonal anti-FliC (Difco ref. 228241) and anti-FljB (Difco ref. 224741) antibodies. Subsequently, the secondary antibody α-rabbit conjugated with Alexa Fluor 488 (Invitrogen) was added. Bacterial DNA was stained with Fluoroshield containing DAPI (4′,6-diamidino-2-pheny-lindole; Sigma). Filaments were imaged using a Zeiss Axio Observer Z1 inverted microscope at ×100 magnification with an Axiocam 506 mono CCD-camera and the Zen 2.6 pro software. The images were further processed with ImageJ.

**Fluorescence microscopy**. Strains harboring a chromosomal FlhA-mNeonGreen fusion were grown in LB until reaching an $OD_{600}$ of 0.7–0.8. Cells were harvested, washed in PBS, and applied on 1% agarose pads (Sigma). Slides were imaged using a Nikon Eclipse Ti2 inverted microscope equipped with a sCMOS Prime BSI (Photometrics) camera, a Lumencor Spectra III light engine (Lumencor), and a CFI Plan Apochromat DM ×60 Lambda oil Ph3/1.40 objective (Nikon) using a LED-DA/FI/TR/Cy5/Cy7-A Filter Cube (Semrock). Z-stacks were acquired and the FlhA-mNeonGreen foci were counted using MicrobeJ[31].

**Blue native PAGE**. Crude membranes were prepared as described in[32] and BN-PAGE was performed as described in[33]. Cells were digested with lysozyme in isotonic buffer, lysed using glass beads, and crude membranes were collected by differential centrifugations. Membrane protein complexes were extracted in 1% *n*-dodecyl-β-D-maltoside and clarified by centrifugation at 20,000 × g. Protein samples were mixed with 1/10th of loading buffer (50% (w/v) glycerol, 250 mM 6-aminocaproic acid, and 5% (w/v) Coomassie Blue G) and loaded on a 3–12% NativePAGE (Thermo Fischer Scientific) and ran according to the manufacturer's instructions. After electrophoresis, the gel was equilibrated in SDS running buffer (25 mM Tris, 0.19 M glycine, and 0.1% SDS) and transferred onto a 0.2 μ PVDF membrane at 30 V for 2 h in transfer buffer (25 mM Tris Base, 192 mM Glycine, and 20% Methanol) supplemented with 0.025% SDS. After transfer, the membrane was washed with methanol to remove residual Coomassie.

**Western blotting**. Membranes were blocked in 5% non-fat milk and probed with the following primary antibodies: monoclonal mouse anti-FLAG (Sigma-Aldrich ref. F1804; dilution 1:3000) and polyclonal rabbit anti-FlhA_C (gift from Tohru Minamino[34], dilution 1:10,000), and the following secondary antibodies: Immun-Star goat anti-rabbit HRP (BioRad ref. 1705046; dilution 1:20,000) and Immun-star goat anti-mouse HRP (Biorad ref. 1705047; dilution 1:20,000). Membranes were developed using Clarity Western ECL (BioRad) and imaged with an ImageQuant LAS 4000 (GE Healthcare).

**Bacterial fitness**. Bacterial fitness was determined from the difference in growth of mutant strains compared to the WT reference. Growth was monitored by measuring every 5 min the optical density of liquid cultures cultivated in 96-well plates at a wavelength of 600 nm using Cytation3 or Synergy H1 plate readers (Biotek).

**Data analysis**. Data were processed using pymol 2.5, ImageJ 1.52i, MicrobeJ, and analyzed using python 3.8 (numpy 1.19.2, pandas 1.1, matplotlib 3.3, seaborn 0.11, scipy 1.5). Protein logos were generated using weblogo 3.7.5. The phylogenic tree of the M-loop mutants was graphed using iTOL 5.7.

**Reporting summary**. Further information on research design is available in the Nature Research Reporting Summary linked to this article.

## Data availability
The coordinates of the FliPQR complex in its closed state are available at the Protein Data Bank under the accession number 6f2d. Biological materials and other data underlying this article are available upon reasonable request to the corresponding authors. Source data are provided with this paper.

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

## Acknowledgements

We would like to thank the Erhardt and Charpentier laboratories for continuous help and support, useful discussions, and for critical comments on the manuscript; Tim Sullivan and Heidi Landmesser for expert technical assistance; Tohru Minamino for providing us with the anti-FlhA_C antibody. This work was supported in part by the European Research Council (ERC) under the European Union's Horizon 2020 research and innovation program (grant agreement no. 864971), the Deutsche Forschungsgemeinschaft (DFG) research grant no. ER 778/2-1 (to M.E.), the Max Planck Society (to E.C.), the German Research Foundation (DFG, Leibniz Prize to E.C.), and a fellowship from the Alexander von Humboldt foundation (to T.T.R.). We acknowledge support by the Open Access Publication Fund of Humboldt-Universität zu Berlin. The funders had no role in study design, data collection and analysis, decision to publish, or preparation of the manuscript.

## Author contributions

T.T.R. and M.E. conceived the project, designed the study, wrote and revised the paper. T.T.R., S.H., M.H., U.v.L., A.G., and D.F.B. performed the experiments. T.T.R., M.E., and D.F.B. analyzed and interpreted the data with input from all authors. E.J.C.G. contributed to the bioinformatic analyses. E.C., M.E., and T.T.R. contributed funding and resources. All authors commented and approved the manuscript.

## Funding

## Competing interests

The authors declare no competing interests.
