## [Peer Review File · Nature Communications]

Editorial Note: Parts of this Peer Review File have been redacted as indicated to remove third-party material where no permission to publish were obtained

REVIEWER COMMENTS

Reviewer #1 (Remarks to the Author):

This manuscript investigates protein secretion from the cytoplasm by the flagellar type III secretion system. The authors are able to build on previous findings that the FliP protein forms the secretion pore with an apparent methionine gasket through which amino acids pass through, and the 3-dimensional structure of the pore complex (FliP-Q-R) has been solved. Using the solved structure and bioinformatic analysis of conserved residues, the authors are able to perform a targeted structure-function analysis of this complex to test the hypothesis that the FliP pentamer forms a methionine gasket with three Met residues donated by each FliP monomer, and FliR serves as a secretion plug. What is remarkable about the EA structure is its ability to assemble above the cytoplasmic membrane. Each of the three methionine residues in FliP were replaced with the other 19 amino acids and the 57 constructs were compared to wild-type (all met at the 3 positions). Remarkably, all constructs formed complexes in the inner membrane at wild-type levels. Using a combination of motility and secretion of a beta-lactamase reporter fused to the hook protein they were able to access the effect of all substitutions on secretion. The finding of the M211R as a suppressor of the M210E was nice. Early substrate secretion correlated perfectly with motility and pore leakage was measured with a clever guanidinium assay to show that leakage was well tolerated with minimal or no effects on motility at the 210 and 211 positions.

Regarding the NNSx3 mutagenesis. Did they screen for the ~33,000 possibilities? I believe that one would need some 300,000 screened to have a ~99% confidence that all were screened. The data look good though so I'm not too concerned. Overall, the experiments in Figure 3 represent an impressive set of data.

The authors go on to characterize FliR as a potential plug. It's a very solid, incredibly thorough and convincing structure-function study.

Figure 5 data analyze leakiness of assembly mutants. It's clear that only the final assemble EA will allow leakage in the ATPase-defective backgrounds.

Overall, an impressive study and quite timely. It's one thing to have a structure. It's quite another to show how it works. This study does that beautifully.

Minor comments

Page 1, line 37: "- up to tens of thousands". Is this more like ~5,000?

Page 1, line 40: change "extreme secretion speed" to something like "high rate of secretion", there is nothing extreme here, it is what evolution selected.

Page 2, line 42: change "but not" to "with minimal or no loss of water or ions from the cytoplasm" – or something to that effect – you do not know that there is/are NO water/ions lost.

Reviewer #2 (Remarks to the Author): or preventing

At first glance, I was immediately excited to read this paper. The figures are beautifully laid out, absolutely gorgeous, and I was primed to believe the data, it looked good skimming it. There was a strong genetic component to disrupt residues putatively involved in plugging the flagellar type III secretion apparatus. I suspect the analysis and results are solid, judging from the figures alone, but the text was so unbelievably poor and so annoying to read I just don't know what to do. Everything about what was done, how it was done, and what the actual results were, was glossed over so quickly with implicit shorthand and jargon and instead most of the paper is pure interpretation ad nauseam in the context of the model. Worse, the text is so overwritten, the paper could be cut in half with ease

and be twice as clear. Just goes on and on without describing anything clearly except the model based largely on the structure. The figure legends, aren't figure legends, they are interpretations. I've never seen anything like this. A massive directed mutagenesis experiment was performed in FliP, only to be completely outshined and superseded by a random approach that told the authors the exact same thing, but better (if I'm understanding correctly because again, nothing was explained that well). After a while, everything seemed like mass without substance. I stopped reading at line 341. Please understand, I'm an ally. I sat down to read and review the paper thinking it was going to be a breeze and instead, it was torture. I would urge the authors consider a full recrafting and rewrite of the paper, where the experiments and the results are so clear that the model only has to be stated once. I suspect the genetic analysis is beautiful but I want to know what was done and what the results were without constant intrusion of the model.

The title doesn't seem to say anything specific.

Lines 74-99 seem to be introductory material, or rather, don't seem to contain new results. Not the end of the world but it was somewhat confusing to me as I was double checking to make sure I didn't miss an experiment. Actually, the intro material extends to line 117 and becomes somewhat repetitive in information.

I believe the screen was for genetic suppressors, and the mutants that came back I believe were pseudorevertants, although I'm not entirely clear on this. What was the starting sequence and what was the pseudorevertant sequence? My definition of pseudorevertant would be mutating one of the methionine to a different residue and the suppressor restoring the methionine using a non-parental codon, the problem being that there is only one codon for methionine and thus they would have to be true revertants if my reading is correct. Otherwise, I believe they may be talking about intra-genic suppression. Not sure.

Paragraph starting line 132. This paragraph seemed to be written out-of-order. For instance the authors say why they use motility as an assay (line 143) after already presenting the results, and even the rationale in line 125. Then the authors end with a mention about confirming the results (which results?) with a hook protein secretion assay but don't describe either the outcomes or how the outcome was confirmatory. Worse, I couldn't actually understand what the result was as substitutions were described as "being tolerated" but tolerated for what? Some could actually be constructed and the rest were lethal?

Line 144. I believe the authors are measuring secretion of a FlgE-bla fusion and not FlgE directly.
Line 146. Could the authors describe how guanidinium tests membrane permeability? Ion leakage measured how? This paragraph seems important to the argument but is so short and so poorly described that I cannot understand what was done or what happened.

Line 154. "The motility and permeability data were combined and clustered by amino acid." I honestly have no idea what this means but the undefined jargon "cluster" is used extensively in the paragraph so I couldn't understand what is going on. In terms of the mutagenesis, I believe that the authors changed each methionine singly to each other amino acid (19 mutants at each position for a total of $19 \times 3 = 57$ mutants). But the results aren't explaining if there are differences at each position or if they all behave the same. And if there are only single mutants, why is a MER double substitution mentioned in line 137. From the text it just isn't clear what was done genetically, and I think this should be very straightforward.

Line 169. "This suggests a structural role for this residue, which is well supported by the structural data." I don't understand this argument, it seems tautological.

Line 180-204 doesn't seem to contain results.

Line 205. What does NNSx3 mutagenic sequence mean? Line 209. Please describe the three different approaches. Line 212 how were functional combinations selected? I imagine that they spotted the pool on a motility plate, harvested from the motile halo, and sequenced the pool but I shouldn't have to imagine how this works. And why is line 212 fronted with "on the contrary" with regard to selection?

Paragraph starting line 253. Again, so poorly described I cannot tell what was done.

Figure legends 4 and 5 describe conclusions drawn from the figures, they do not describe the experiment (what kind of experiment it is or what is on the Y-axis). Actually, all of the figure legends are like this.

Based on the setup in the introduction, about serving as a plug for small molecules, I would have expected a stronger growth defect in Fig 7 B, and salt independence in Fig 4B. Can a FliP M-loop mutant be combined with an R-plug FliR mutant and be viable?

Signed, Dan Kearns

Reviewer #3 (Remarks to the Author):

This manuscript by Thibaud Renault deals with the functional analysis of gating of the core export apparatus of bacterial flagella. Towards this, they focus on a highly conserved region of methionines in FliP, located at the entrance of the pore of this complex, and on a central plug formed by the protein FliR.

The authors performed a very deep genetic analysis and classified the observed mutants based on their performance in flagellar function (swimming), basic type III secretion function and leakiness of the pore.

The authors conclude that the conserved methionines form a deformable gasket that seals the pore during the high-speed export of flagellar subunits in cooperation with the FliR-plug. The presented data also indicate that pore gating relates to the stabilization of the core export apparatus by conserved intermolecular salt bridges in close proximity to the M-gasket.

The manuscript is very well written and illustrated. The methodology is sound and described to a sufficient detail. Overall, this beautiful analysis provides the first deeper functional analysis of the type III secretion system core export apparatus since presentation of its structure. The results elucidate a critical aspect of these molecular machines and help to understand basic biological principles (sealing and gating of protein export across biological membranes).

I only have a few critical points that I list point by point below.

1. The authors hypothesize that charged residues and tryptophan substituting methionines of the gasket impair pore assembly or stability. This hypothesis is rather easy to test by performing blue native PAGE followed by Western blotting and immunodetection of the FliP-FLAG as the authors performed previously (Fabiani 2017). It would be nice to show this at least for a few exemplary mutants as it would have implications for the interpretation of the data if the mutants unexpectedly turn out to assemble correctly.

2. The authors find that the substitution of two methionines by opposing charges results in a neutral effect and they take this as support that a net charge is detrimental for FT3SS function. This interpretation may be a bit far-stretched as not function but biogenesis of the mutants may be affected. It is still believed that FliPQR are inserted into the membrane at the Sec-translocon prior to complex assembly. Therefore, the authors should at least bioinformatically check the effect of net

charges in this "cytoplasmic" loop on the topogenesis of FliP. A screwed "transmembrane" topology would already explain a substantial amount of the the findings but has nothing to do with the make up of the gasket itself. In fact, the data in Suppl. Fig. 2A show that substitutions of the second or third methionine with charged residues result in a substantial reduction of the accumulation levels of FliP, indicating a stability problem, possibly caused by screwed biogenesis.

3. I feel that the authors over-interpret their findings that only mutations in the "dispensable" FliH/I/J ATPase caused leakage (LL 330 ff) by concluding that active secretion is required for leakiness. See also statement in lines 504/505.

Firstly, the non-leaky mutants (Δ fliB, Δ fliA, Δ fliF) are unlikely to assemble the FliPQR core complex into the flagellar basal body, meaning these mutants do not allow to independently judge leakiness in a "non-active secretion" state (by the way, how does the stability and complex assembly of FliP perform in these mutants?).

Secondly, Δ fliHIJ and Δ fliI mutants are very poorly secreting, much less then the 40% reduction in leakiness, so the two phenotypes do not seem to correlate well.

It rather looks to me that one needs the FliPQR complex positioned correctly relative to the membrane, in the context of the other transmembrane domains of FliA and FliB, to be able to gate and become leaky and that this has little to do with active secretion. It would be informative to see how the leakiness assay performs in the MAA mutant (and otherwise wild type) when the cells are treated with CCCP, thus abrogating possible ongoing secretion during the course of this assay.

4. Suppl. Fig. 2: The legends of panels C-E lack a explanation of "-" and "+". I suppose, Δ fliP and wt.

Samuel Wagner

Response to reviewers

We would like to thank the three reviewers for their expertise and their rigorous reading of our manuscript. We are thankful for their feedback and suggestions and for their constructive criticism that helped us to improve our manuscript.

We are confident that we were able to address all the comments. Changes in the manuscript are highlighted in yellow to facilitate reading and a point-to-point response is provided below.

Reviewer #1 (Remarks to the Author):

This manuscript investigates protein secretion from the cytoplasm by the flagellar type III secretion system. The authors are able to build on previous findings that the FliP protein forms the secretion pore with an apparent methionine gasket through which amino acids pass through, and the 3-dimensional structure of the pore complex (FliP-Q-R) has been solved. Using the solved structure and bioinformatic analysis of conserved residues, the authors are able to perform a targeted structure-function analysis of this complex to test the hypothesis that the FliP pentamer forms a methionine gasket with three Met residues donated by each FliP monomer, and FliR serves as a secretion plug. What is remarkable about the EA structure is its ability to assemble above the cytoplasmic membrane. Each of the three methionine residues in FliP were replaced with the other 19 amino acids and the 57 constructs were compared to wild-type (all met at the 3 positions). Remarkably, all constructs formed complexes in the inner membrane at wild-type levels. Using a combination of motility and secretion of a beta-lactamase reporter fused to the hook protein they were able to access the effect of all substitutions on secretion. The finding of the M211R as a suppressor of the M210E was nice. Early substrate secretion correlated perfectly with motility and pore leakage was measured with a clever guanidinium assay to show that leakage was well tolerated with minimal or no effects on motility at the 210 and 211 positions.

Regarding the NNSx3 mutagenesis. Did they screen for the ~33,000 possibilities? I believe that one would need some 300,000 screened to have a ~99% confidence that all were screened. The data look good though so I'm not too concerned. Overall, the experiments in Figure 3 represent an impressive set of data.

RE: This is correct. We have not isolated all functional combinations as we only have selected a bit more than 300 functional variants (Suppl. Fig. 3B), which corresponds to 254 unique motifs. However, as we used a degenerate oligonucleotide for the recombination, all ~33,000 combinations were included in the selections/screening with a high confidence. Yet, we believe that it is not needed to be exhaustive to understand the required physicochemical properties of the M-loop residues. The goal of this experiment was to challenge a given amino-acid at a given position in an otherwise random context to identify the physicochemical properties that enabled function. Our sample allowed us to test on average 20 combinations per amino-acid and per position and, importantly, we identified a clear difference between the selected and unselected conditions (Fig. 3C and Suppl. Fig. 3B).

Furthermore, while they are ~33,000 combinations of codons (32^3), there are less combinations of amino-acids ($20^3 = 8,000$). By comparing Lac⁺ to Lac⁻ clones, we estimate that only ~9% of the combinations have a minimal level of functionality as reported by our FliB::lacZ assay (Suppl. Fig. 3C). Of note, when we selected for functional motifs that resulted in motility, we indeed obtained one MMM variant despite the expected probability of only ~1/33,000.

The authors go on to characterize FliR as a potential plug. It's a very solid, incredibly thorough and convincing structure-function study.

Figure 5 data analyze leakiness of assembly mutants. It's clear that only the final assemble EA will allow leakage in the ATPase-defective backgrounds.

Overall, an impressive study and quite timely. It's one thing to have a structure. It's quite another to show how it works. This study does that beautifully.

RE: Thank you very much for your thoughtful comments.

Minor comments

Page 1, line 37: “- up to tens of thousands”. Is this more like ~5,000?

RE: This is still a debated question, however, we have several indications that the maximal secretion speed is in the range of tens thousands of amino-acids per second. As an example, Schlumberger *et al.* have used time-lapse microscopy to estimate that in the case of the vT3SS of *Salmonella*, ~6,000 molecules of SipA (685 amino-acids) are secreted in ~100–600 seconds, which gives a range of speed of ~7,000–41,000 aa/s (<https://doi.org/10.1073/pnas.0503407102>). We have ourselves estimated the initial secretion speed of flagellin via the fT3SS in *Salmonella* to be of ~30 molecules per second (~15,000 aa/s) (<https://doi.org/10.7554/elife.23136>). By improving our filament labeling protocol published in Renault *et al.* eLife 2017, we could recently measure a secretion speed of ~10,000 aa/s and estimated a maximum speed of ~20,000 aa/s (unpublished).

Page 1, line 40: change “extreme secretion speed” to something like “high rate of secretion”, there is nothing extreme here, it is what evolution selected.

RE: We thank the reviewer for this suggestion, which has been incorporated in the text.

Page 2, line 42: change “but not” to “with minimal or no loss of water or ions from the cytoplasm” – or something to that effect – you do not know that there is/are NO water/ions lost.

RE: We thank the reviewer for this suggestion, which has been incorporated in the text.

Line 263: change to: while minimizing or preventing

RE: We thank the reviewer for this suggestion, which has been incorporated in the text.

Reviewer #2 (Remarks to the Author):

At first glance, I was immediately excited to read this paper. The figures are beautifully laid out, absolutely gorgeous, and I was primed to believe the data, it looked good skimming it. There was a strong genetic component to disrupt residues putatively involved in plugging the flagellar type III secretion apparatus. I suspect the analysis and results are solid, judging from the figures alone, but the text was so unbelievable poor and so annoying to read I just don't know what to do. Everything about what was done, how it was done, and what the actual results were, was glossed over so quickly with implicit shorthand and jargon and instead most of the paper is pure interpretation ad nauseam in the context of the model. Worse, the text is so overwritten, the paper could be cut in half with ease and be twice as clear. Just goes on and on without describing anything clearly except the model based largely on the structure. The figure legends, aren't figure legends, they are interpretations. I've never seen anything like this. A massive directed mutagenesis experiment was performed in FliP, only to be completely outshined and superseded by a random approach that told the authors the exact same thing, but better (if I'm understanding correctly because again, nothing was explained that well). After a while, everything seemed like mass without substance. I stopped reading at line 341. Please understand, I'm an ally. I sat down to read and review the paper thinking it was going to be a breeze and instead, it was torture. I would urge the authors consider a full recrafting and rewrite of the paper, where the experiments and the results are so clear that the model only has to be stated once. I suspect the genetic analysis is beautiful but I want to know what was done and what the results were without constant intrusion of the model.

RE: Thank you for taking the time to review our manuscript and communicating your honest opinion. We very much appreciate your feedback.

We have taken great care to try improving the clarity of the manuscript by more explicitly stating the rationale of our experiments and explaining better the used methodology and we believe that the revised manuscript is much improved.

The title doesn't seem to say anything specific.

RE: We investigated in our study multiple features of the fT3SS secretion pore that, according to our results, together are required in order to preserve the membrane barrier during high speed protein translocation. Therefore, we feel that we would omit some important aspects of our findings if we would change the title to a more explicit statement. However, we are open to suggestions if a more specific title is deemed more appropriate.

Lines 74-99 seem to be introductory material, or rather, don't seem to contain new results. Not the end of the world but it was somewhat confusing to me as I was double checking to make sure I didn't miss an experiment. Actually, the intro material extends to line 117 and becomes somewhat repetitive in information.

RE: It is true that this section does not contain new experimental data *per se*. We did, however, re-analyze published data to illustrate the organization of the FliPQR complex and the structural homology between the different sub-units. We believe that this information is important to familiarize with the proteins and domains that are studied in this work, and to appreciate their position in the context of the structure.

I believe the screen was for genetic suppressors, and the mutants that came back I believe were pseudorevertants, although I'm not entirely clear on this. What was the starting sequence and what was the pseudorevertant sequence? My definition of pseudorevertant would be mutating one of the

methionine to a different residue and the suppressor restoring the methionine using a non-parental codon, the problem being that there is only one codon for methionine and thus they would have to be true revertants if my reading is correct. Otherwise, I believe they may be talking about intra-genic suppression. Not sure.

RE: We apologize for the confusing nomenclature. We indeed referred to suppressors and not revertants in the manuscript. We have reworked the text to use the term “suppressor”.

Paragraph starting line 132. This paragraph seemed to be written out-of-order. For instance the authors say why they use motility as an assay (line 143) after already presenting the results, and even the rationale in line 125. Then the authors end with a mention about confirming the results (which results?) with a hook protein secretion assay but don't describe either the outcomes or how the outcome was confirmatory. Worse, I couldn't actually understand what the result was as substitutions were described as “being tolerated” but tolerated for what? Some could actually be constructed and the rest were lethal?

RE: We have reworked the text to improve its clarity. We now describe the methods in the previous section and made it explicit that the reported parameter is flagellar function (motility or secretion).

Line 144. I believe the authors are measuring secretion of a FlgE-bla fusion and not FlgE directly.

RE: We apologize for the confusion, it is indeed a FlgE-Bla fusion as described in the methods. We have amended the main text to make it more explicit.

Line 146. Could the authors describe how guanidinium tests membrane permeability? Ion leakage measured how? This paragraph seems important to the argument but is so short and so poorly described that I cannot understand what was done or what happened.

RE: We have improved the methods section and reference our previous work in which the assay is described in detail.

Briefly, this method relies on monitoring the changes in OD600 resulting from the water movements across the membrane. The initial transfer into high-salt media causes rapid water efflux that increases refractility of the cells and causes a rapid increase in OD 600 (typically by about 10–15%). This is largely complete within a few seconds. In a subsequent slower phase, optical density decreases as water re-enters the cell, at a rate that is mainly dictated by re-entry of the osmotic agent.

Line 154. “The motility and permeability data were combined and clustered by amino acid.” I honestly have no idea what this means but the undefined jargon “cluster” is used extensively in the paragraph so I couldn't understand what is going on. In terms of the mutagenesis, I believe that the authors changed each methionine singly to each other amino acid (19 mutants at each position for a total of $19 \times 3 = 57$ mutants). But the results aren't explaining if there a differences at each position or if they all behave the same.

RE: We apologize for the insufficient clarity and have reworked the text to replace the term cluster by group. We have used the membrane barrier (leakage) and motility function scores to classify amino-acids and regroup those that led to similar phenotypes (those groups are called

“clusters”). This was achieved using python and the seaborn library (that itself relies on the scipy.cluster library).

We used the same approach in Figure 3F for which we now have included a summary in the Methods section.

And if there are only single mutants, why is a MER double substitution mentioned in line 137. From the text it just isn't clear what was done genetically, and I think this should be very straightforward.

RE: This mutant is a suppressor of the FliP M210E variant that spontaneously regained motility in soft agar plates (Figure S2G and Table S1).

Line 169. “This suggests a structural role for this residue, which is well supported by the structural data.” I don't understand this argument, it seems tautological.

RE: We meant here that our biochemical analysis suggested a structural role of the residue M209, in opposition to the functional role in the maintenance of membrane barrier that is supported by residues M210 and M211. We actually obtained this data before the structure of the pore was known, and the structural data confirmed our hypothesis. We have reworked the text to improve clarity.

Line 180-204 doesn't seem to contain results.

RE: This paragraph summarizes the results of the single mutagenesis approach in order to better highlight the difference with the triple and random mutagenesis approaches that are described afterwards.

Line 205. What does NNSx3 mutagenic sequence mean?

RE: This represents the three degenerate codons that have been used for the random mutagenesis. Classically, N indicates any of A/C/G/T nucleotides and S any of G/C. We have updated the main text and methods to make this explicit.

Line 209. Please describe the three different approaches.

RE: We have added a summary of the methods in the main text:

Briefly, we first quantified the motility of mutants using soft-agar swim plates as a proxy for functional flagellar assembly. Second, we quantified secretion of the early fT3SS substrate FlgE (hook protein) fused to beta-lactamase TEM-1 as a proxy of the protein secretion capability of the fT3SS (10). Third, we quantified gene expression from a flagellar class III promoter using a transcriptional lac operon fusion to fljB as a proxy of hook-basal-body completion (gene expression from class III promoters is dependent on secretion of the anti-sigma factor FlgM after hook completion).

Also, the assays are described in the methods and illustrated in Suppl. Fig. 1C and Suppl. Fig. S3A.

Line 212 how were functional combinations selected? I imagine that they spotted the pool on a motility plate, harvested from the motile halo, and sequenced the pool but I shouldn't have to imagine how this works. And why is line 212 fronted with "on the contrary" with regard to selection?

RE: We have used three methods (selection by motility, FlgE-bla secretion coupled with selection on ampicillin plates, screening of functional ft3SS using a FljB::LacZ reporter) to isolate functional M-loop mutants from the pool of random motifs. In parallel, we also sequenced all mutants obtained in a plate without selection as a control (thus the "on the contrary"). Those two conditions (unselected and selected for T3SS function) are presented in the heatmaps of Figure 3C, which we have edited to improve clarity. We also have added a section in the methods, and the method is illustrated in Suppl. Fig. S3A.

Paragraph starting line 253. Again, so poorly described I cannot tell what was done.

RE: We now have improved the text and added a section in the methods. Briefly, we used clustering analysis to classify amino-acid properties that are positively or negatively correlated with motility function (Figure 3F). For this, we have used the random variants of the M-loop that were selected for T3SS function and individually quantified their motility in soft agar to determine a relative motility score (0 to 100%). For each tested motif, we translated each amino-acid to numerical values using amino-acid scales that represent various physico-chemical properties (cf. Methods), and computed a matrix of correlations between amino-acid properties and motility function scores. Finally, we applied the classification algorithm to regroup the similar/closest data points.

Figure legends 4 and 5 describe conclusions drawn from the figures, they do not describe the experiment (what kind of experiment it is or what is on the Y-axis). Actually, all of the figure legends are like this.

RE: In a large part of our manuscript, we used three functional assays to characterize function of the ft3SS (motility, secretion, and flagellum assembly), and one assay to report water movement through the pore. Those assays are described early in the main text and are illustrated in Figure S1C (and Figure S3A when they were used for selection). Those assays were used repetitively throughout the manuscript and thus we did not want to repeat this information in all the figures. Nevertheless, we now have reworked the figures and figure legends to improve clarity.

Based on the setup in the introduction, about serving as a plug for small molecules, I would have expected a stronger growth defect in Fig 7 B, and salt independence in Fig 4B. Can a FliP M-loop mutant be combined with an R-plug FliR mutant and be viable?

RE: Yes, those two mutations can be combined and are viable (Figure 4E). We show however that leakage through the T3SS induces a loss of fitness (Figure 7).

Signed, Dan Kearns

Reviewer #3 (Remarks to the Author):

This manuscript by Thibaud Renault deals with the functional analysis of gating of the core export apparatus of bacterial flagella. Towards this, they focus on a highly conserved region of methionines in FliP, located at the entrance of the pore of this complex, and on a central plug formed by the protein FliR.

The authors performed a very deep genetic analysis and classified the observed mutants based on their performance in flagellar function (swimming), basic type III secretion function and leakiness of the pore.

The authors conclude that the conserved methionines form a deformable gasket that seals the pore during the high-speed export of flagellar subunits in cooperation with the FliR-plug. The presented data also indicate that pore gating relates to the stabilization of the core export apparatus by conserved intermolecular salt bridges in close proximity to the M-gasket.

The manuscript is very well written and illustrated. The methodology is sound and described to a sufficient detail. Overall, this beautiful analysis provides the first deeper functional analysis of the type III secretion system core export apparatus since presentation of its structure. The results elucidate a critical aspect of these molecular machines and help to understand basic biological principles (sealing and gating of protein export across biological membranes).

RE: Thank you very much for your helpful and constructive review and comments on our manuscript.

I only have a few critical points that I list point by point below.

1. The authors hypothesize that charged residues and tryptophan substituting methionines of the gasket impair pore assembly or stability. This hypothesis is rather easy to test by performing blue native PAGE followed by Western blotting and immunodetection of the FliP-FLAG as the authors performed previously (Fabiani 2017). It would be nice to show this at least for a few exemplary mutants as it would have implications for the interpretation of the data if the mutants unexpectedly turn out to assemble correctly.

RE: We thank the reviewer for his suggestion. We now have performed BN-PAGE for a few mutants. Our results show that increased flexibility in the M-loop (AAA mutant), and charges (M211R, M210E, EGR mutants) partially impair complex formation (FigureS2B). This is not the case for a deletion of the C-terminal domain of FlhA nor for the M210A mutant.

2. The authors find that the substitution of two methionines by opposing charges results in a neutral effect and they take this as support that a net charge is detrimental for β T3SS function. This interpretation may be a bit far-stretched as not function but biogenesis of the mutants may be affected. It is still believed that FliPQR are inserted into the membrane at the Sec-translocon prior to complex assembly. Therefore, the authors should at least bioinformatically check the effect of net charges in this “cytoplasmic” loop on the topogenesis of FliP. A screwed “transmembrane” topology would already explain a substantial amount of the the findings but has nothing to do with the make up of the gasket itself. In fact, the data in Suppl. Fig. 2A show that substitutions of the second or third methionine with charged residues result in a substantial reduction of the accumulation levels of FliP, indicating a stability problem, possibly caused by screwed biogenesis.

RE: The mutations that we introduced in the M-loop do not change the topology that is predicted using the TOPCONS prediction program. It does, however, affect the ΔG in the M-loop region (see below), but it remains difficult to predict structural changes only from this value. Furthermore, the AAA mutant, which assembles the flagellum inefficiently, has a signal almost identical to that of the WT, suggesting that the ΔG value of the M-loop alone is not a good predictor of stability.

It is possible that only subtle structural changes affect stability of the export apparatus. To address this, we investigated a FlhA-mNeonGreen reporter and quantified the number of foci that are assembled in the membrane. It has previously been shown that proper assembly of FlhA requires FliPQR (10.1111/mmi.12529). Using this method we tested stability of the export apparatus for single mutants to alanine, tryptophane, or charges. We observed that most charged mutants are unstable, as well as the tryptophane mutant of the first methionine and mutants with increased M-loop stability (AAA, MAA).

3. I feel that the authors over-interpret their findings that only mutations in the “dispensable” FliH/I/J ATPase caused leakage (LL 330 ff) by concluding that active secretion is required for leakiness. See also statement in lines 504/505.

Firstly, the non-leaky mutants ($\Delta flhB$, $\Delta flhA$, $\Delta fliF$) are unlikely to assemble the FliPQR core complex into the flagellar basal body, meaning these mutants do not allow to independently judge leakiness in a “non-active secretion” state (by the way, how does the stability and complex assembly of FliP perform in these mutants?).

RE: We agree that full deletions of *flhB* and *flhA* could result in an assembly defect. Although, in Kuhlen *et al.* Nature Communications 2020, it actually seems that FlhA was not expressed and yet the complex is properly assembled, and in a closed state.

Yet, we would like to emphasize that we also have tested mutants of the cytoplasmic domains of the (FlhA- ΔC -term and FlhB- ΔC -term), which are expected to properly assemble the FliPQR complex (Abrusci *et al.* NSMB, 2013). We confirmed this using BN-PAGE for the FlhA- ΔC -term mutant (Figure S2B).

Secondly, $\Delta fliHIJ$ and $\Delta fliI$ mutants are very poorly secreting, much less than the 40% reduction in leakiness, so the two phenotypes do not seem to correlate well.

RE: We agree that the interpretation of the phenotype is not obvious, however, we believe that our hypothesis is currently the most rationale option to explain this result and we provide the following explanation:

– First, FliHIJ play no role in export apparatus assembly. Thus, the reduced leakage phenotype that we observed in the absence of the FliI ATPase or with the FliI mutant that is impaired in

ATP hydrolysis (Figure 5B) cannot be attributed to an assembly defect. The other options are (a) a major general defect in secretion, (b) a defect in priming for secretion.

– It is well established that the FliH/I/J proteins are crucial to enable efficient secretion, flagellar assembly, and motility, nevertheless they are “dispensable” in the sense that secretion and even complete flagellum assembly can occur – inefficiently – in their absence. For instance, some residual motility and filament assembly was reported in the Δ fliH/I/J mutant by Paul *et al.* (left figure thereafter).

– Furthermore, it is also believed that there is not a linear/proportional relationship between ATP Hydrolysis and function, as demonstrated by Minamino *et al.* (right figure thereafter). The three conditions in the Minamino study, WT FliI, E211D (100× reduced ATPase activity), and E211Q (no ATP hydrolysis), reveal that secretion and motility are not linearly correlated with ATP hydrolysis, and with each other. They suggest an “ignition-key” mechanism in which infrequent ATP hydrolysis is sufficient to trigger pore opening and substrate secretion. (Redacted)

Similarly, we show in our manuscript that leakage increases with increasing ATPase function. The most affected conditions (Δ fliH/I/J, Δ fliI, and FliI E211Q) have a “basal” level of leakage, and the E211D mutant that has a secretion level comparable to that of the WT ATPase also displays a high level of leakage in presence of the FliP MAA mutation. To this extent, we believe that our result demonstrate that active secretion (*i.e.*, at least a minimal level of secretion *e.g.* triggering pore opening) is required to induce leakage, and the levels of secretion and of leakage are correlated (although not linearly). We now provide additional data suggesting that leakage is in fact proportional to secretion (see next response).

Finally, we do not see a reason why leakage should be linearly proportional to secretion. We can easily imagine that slowly secreting substrates or discontinuous secretion events are more prone to facilitate leakage in the context of a deficient M-gasket. We actually had previously demonstrated that a trapped substrate compensated the growth defect induced by the FliP Δ M in presence of the large choline ion but not of cadmium, which indicates that leakage through the

FT3SS is not an all-or-nothing mechanism, but rather a complex phenomenon that is affected by several parameters, including the position of the secreted substrate relative to the export apparatus.

It rather looks to me that one needs the FliPQR complex positioned correctly relative to the membrane, in the context of the other transmembrane domains of FlhA and FlhB, to be able to gate and become leaky and that this has little to do with active secretion. It would be informative to see how the leakiness assay performs in the MAA mutant (and otherwise wild type) when the cells are treated with CCCP, thus abrogating possible ongoing secretion during the course of this assay.

RE: We now have performed the suggested experiment (Figure 5C). We measured leakage in the MAA mutant (using the WT MMM and Δ fliP as controls) in presence of various concentrations of CCCP. We observed that there is a correlation between secretion and leakage suggesting that leakage can only occur during active secretion. For the highest concentration of CCCP, in which secretion of flagellin is barely detected, the level of leakage is similar to that of the non-leaky controls.

4. Suppl. Fig. 2: The legends of panels C-E lack a explanation of “-“ and “+”. I suppose, Δ fliP and wt.

RE: Thank you for pointing this oversight. “-“ and “+” indeed refer to Δ fliP and wild-type (MMM), which is now mentioned in the figure legend.

Samuel Wagner

REVIEWERS' COMMENTS

Reviewer #1 (Remarks to the Author):

Great work, great re-write, all my concerns were addressed. There are just a couple of typos noticed:

line 121, italicize "lac"

line 266, "NNS" seems a different font

line 442, change "(pmf)" to "(PMF)"

Reviewer #2 (Remarks to the Author):

I'm grateful to the authors for taking my concerns seriously, despite my poor attitude. They have improved the clarity of the work and they have addressed my comments.

Reviewer #3 (Remarks to the Author):

The authors have responded thoroughly to my critique and addressed it to satisfaction. I feel that the manuscript has improved and that the findings are very clear. I have no further comments.